# Are Object-Centric Representations Better at Compositional Generalization?

**Ferdinand Kapl** [1 2] **Amir Mohammad Karimi Mamaghan** [3] **Maximilian Seitzer** [4] **Karl Henrik Johansson** [3] **Carsten Marr** [5] **Stefan Bauer** [1 2 6] **Andrea Dittadi** [1 2 4]

## Abstract

Compositional generalization, the ability to reason about novel combinations of familiar concepts, is fundamental to human cognition and a critical challenge for machine learning. Object-centric (OC) representations, which encode a scene as a set of objects, are often argued to support such generalization, but systematic evidence in visually rich settings is limited. We introduce a Visual Question Answering benchmark across three controlled visual worlds (CLEVRTex, Super-CLEVR, and MOVi-C) to measure how well vision encoders, with and without object-centric biases, generalize to unseen combinations of object properties. To ensure a fair and comprehensive comparison, we carefully account for training data diversity, sample size, representation size, downstream model capacity, and compute. We use DINOv2 and SigLIP2, two widely used vision encoders, as the foundation models and their OC counterparts. Our key findings reveal that (1) OC approaches are superior in harder compositional generalization settings; (2) original dense representations surpass OC only on easier settings and typically require substantially more downstream compute; and (3) OC models are more sample efficient, achieving stronger generalization with fewer images, whereas dense encoders catch up or surpass them only with sufficient data and diversity. Overall, object-centric representations offer stronger compositional generalization when any one of dataset size, training data diversity, or downstream compute is constrained.

[1]Technical University of Munich [2]Helmholtz AI, Munich [3]KTH Royal Institute of Technology [4]MPI for Intelligent Systems, Tübingen [5]Institute of AI for Health, Computational Health Center, Helmholtz Munich [6]Munich Center for Machine Learning (MCML). Correspondence to: Ferdinand Kapl <ferdinand.kapl@tum.de>.

*Proceedings of the 43rd International Conference on Machine Learning*, Seoul, South Korea. PMLR 306, 2026. 

## 1. Introduction

Compositionality, the ability to perceive and generalize to novel combinations of familiar elements, is widely seen as a cornerstone of human cognition and has long been linked to the systematic ability of humans to understand and produce novel expressions from known parts (Fodor & Pylyshyn, 1988; Treisman, 1996; Chomsky, 2002). In machine learning, *compositional generalization*—the robustness of models to novel combinations of familiar concepts—has been explored in various forms. In natural language, compositional generalization can be assessed by testing the response of the model to rearranged or recombined words and numbers (Lake & Baroni, 2018; Dziri et al., 2024); in vision, it may involve creating novel objects by recombining seen object properties or combining known objects in novel configurations (Kim et al., 2024; Haramati et al., 2024; Montero et al., 2024; Abbasi et al., 2024). In practice, multiple studies show brittleness in compositional generalization. VLMs struggle with hard negatives (Huang et al., 2024; Hsieh et al., 2024), and text-to-image models degrade as prompts combine more entities (Wu et al., 2024; Li et al., 2024). Scaling alone has not solved this: accuracy still drops on unseen combinations, and performance depends on pretraining frequency and diversity (Kempf et al., 2025; Wiedemer et al., 2025; Uselis et al., 2025).

These observations have motivated research into vision representations that support compositional generalization more naturally. In particular, *object-centric (OC) representations* represent a scene as a collection of objects, binding different objects to separate *slot vectors* (Locatello et al., 2020; Greff et al., 2020). Because such representations match the natural structure of a scene by decomposing it into discrete objects, they are conjectured to provide more compositional and generalizable representations (Greff et al., 2020; Locatello et al., 2020; Dittadi et al., 2022; Jiang et al., 2023; Brady et al., 2023). However, beyond a few preliminary indications (Yoon et al., 2023; Montero et al., 2024; Kim et al., 2024; Haramati et al., 2024), the relationship between OC representations and compositionality remains largely untested in a systematic and principled manner. In this work, we investigate those claims in greater depth. Specifically, we study how well different visual representations support compositional generalization of object properties on the

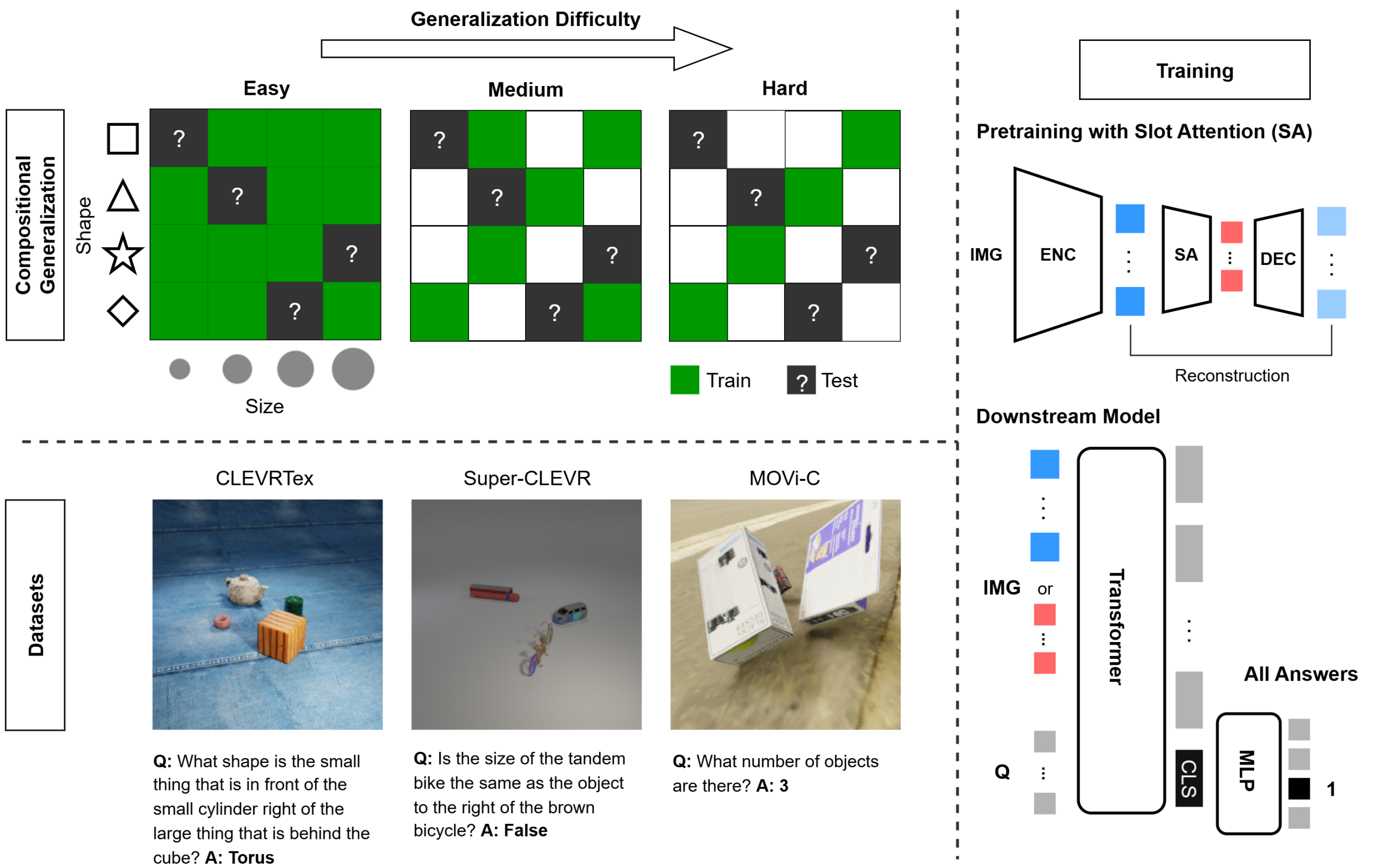

*Figure 1.* **Compositional Generalization.** To increase generalization difficulty, we decrease the number of unique object property combinations that are seen during training. In the conceptual example, each object is defined by its shape and size, which coincides with MOVi-C. **Datasets.** For each generalization difficulty and base dataset, we generate images and corresponding question–answer pairs by sampling objects with the allowed combinations. **Training.** We pretrain object-centric (OC) models by reconstructing the self-supervised (Dense) features from pretrained vision encoders. For VQA downstream training, we concatenate the image features (OC: red; Dense: blue) with the fixed text embeddings and train transformer models of various sizes to predict the answer given image and question.

visual question answering (VQA) task.

The flavor of compositionality we are interested in is *object property composition* (Johnson et al., 2017; Abbasi et al., 2024; Montero et al., 2024; Kim et al., 2024)—the ability of a model to generalize to novel combinations of previously seen object properties. For example, a model trained only on red cubes and blue spheres should be able to successfully handle blue cubes at test time. Because this form of compositionality requires precise control over the factors of variation in the visual world, most works rely on synthetically generated images from a computer graphics tool (Kim et al., 2024; Montero et al., 2024) or a pretrained generative model (Abbasi et al., 2024). Although compositionality is often described as a core motivation for OC representations, its evaluation is typically limited to changing the number of objects (Johnson et al., 2017; Locatello et al., 2020; Karazija et al., 2021; Biza et al., 2023). The works most similar to ours are Kim et al. (2024) and Montero et al. (2024), both investigating compositional generalization of object properties. However, Kim et al. (2024) use a protocol that only allows evaluation of generative models rather than general image representations and do not isolate which design choices contribute to better performance, while Montero et al. (2024) examine compositionality only under the more limited setting of simpler images with a single object. At the same time, several studies suggest that conclusions drawn from visually simple synthetic worlds may not transfer: OC approaches that perform well in such settings can degrade substantially under complex textures, corruptions, or real-world imagery (Karazija et al., 2021; Papa et al., 2022; Seitzer et al., 2022; Drenkow & Unberath, 2024). Together, these gaps motivate a benchmark that targets object property composition while remaining both visually rich and fully controlled.

In order to rigorously study the compositional generalization capabilities of visual representations for object property composition in such visually rich but controlled worlds, we design our own benchmark. Inspired by Kim et al. (2024), we generate images in a CLEVRTex (Karazija et al., 2021), Super-CLEVR (Li et al., 2023), and MOVi-C (Greff et al.,

2022) style, allowing us to precisely define the entire visual world. Specifically, we consider every combination of individual factors of variation (e.g., shape, material, and size) characterizing each object. Then, we reserve 20% of these object combinations for testing compositional generalization while allocating the rest to progressively smaller subsets for training. This ensures that no test objects of the compositional generalization dataset were encountered during training, even though their individual properties were. To evaluate this compositional generalization via VQA, we follow Mamaghan et al. (2025) and Li et al. (2023) by generating question–answer pairs for all images. For each of the three base datasets introduced above, we thus have three different training sets—which we call easy, medium, and hard—and a fixed test set dedicated to the evaluation of compositional generalization, called COOD.

For our comparisons, we focus on pretrained foundation models and OC models that incorporate such foundation models as backbones, a leading approach in this domain. Specifically, we use DINOv2 (Oquab et al., 2024) and SigLIP2 (Tschannen et al., 2025) as the foundation models with DINOSAURv2 (Seitzer et al., 2022; Didolkar et al., 2024) and SigLIPSAUR2 as their OC counterparts. To ensure a fair and comprehensive comparison, we account for differences in representation format by controlling for image representation sizes, both for the number of tokens and the token dimension, ensuring that differences in compute allocation do not unfairly advantage one approach over another. We evaluate all models by training distinct downstream models for the VQA task on training sets of increasing difficulty, testing on in-distribution (ID) as well as compositional out-of-distribution (COOD) generalization sets. Following the framework of Mamaghan et al. (2025), we vary the size of the downstream model and, additionally, the input size of the image representation. Finally, by carefully controlling the visual combinations that models are exposed to at train and test time, we can systematically adjust the hardness of the generalization task until even an oracle with access to ground-truth inputs struggles to generalize at test time.

Our main contributions can be summarized as follows:

- **A controlled benchmark for object-property compositional generalization.** We introduce a VQA benchmark spanning three synthetic visual worlds (CLEVRTex, Super-CLEVR, MOVi-C) with explicit control over object property combinations. Test sets contain held-out combinations of seen properties, enabling systematic evaluation of compositional out-of-distribution (COOD) generalization under varying levels of training diversity (Fig. 1).
- **A fair and systematic comparison of dense and object-centric representations.** We evaluate pretrained foundation models and their OC counterparts under matched representation sizes, downstream model capacity, and FLOPs, *isolating the effect of representation structure* on compositional generalization.
- **Object-centric representations generalize better compositionally under limited data, diversity, or compute.** Across dataset size, training diversity, and downstream compute budgets, OC representations consistently achieve better compositional generalization whenever any one of diversity, data, or compute is constrained. Dense representations match or surpass them only on easier settings with sufficiently large and diverse data and larger downstream models.

## 2. Related Work on Compositionality

This section briefly summarizes different ways in which compositionality has been defined and tested.

**Text.** Lake & Baroni (2018) studied compositionality by training a model to decode natural-language commands into action sequences that feature novel combinations of concepts at test time. Dziri et al. (2024) demonstrated that transformers can fail catastrophically on seemingly simple tasks (e.g., multi-digit integer multiplication) when test conditions differ slightly from training (e.g., more digits).

**Images.** Kim et al. (2024) explored compositionality without language annotations by constructing a visual world of objects with simple attributes (e.g., shape, texture). They controlled which portion of the combinatorial attribute space was shown during training and formulated a generative task where the model must learn and apply transformation rules (e.g., swapping shapes) to unseen combinations at test time. Haramati et al. (2024) probe, among other things, the compositional generalization of different components of their architecture in a reinforcement learning task that involves arranging objects on a table with a robotic arm.

**Text-to-image.** Some recent work frames compositionality as a text-to-image generation task, prompting models with increasingly complex combinations of visual concepts to test that all mentioned concepts appear in the generated image (Wu et al., 2024; Li et al., 2024).

**Image-to-text and VQA.** The *SugarCrepe* benchmark evaluates compositional comprehension by presenting an image alongside a correct caption and a closely matched "hard negative", which can involve object swapping or replacement (Hsieh et al., 2024). The model must choose the caption that accurately describes the image, extending earlier approaches such as Ma et al. (2023).

**Object-centric representations.** In the context of reinforcement learning, Yoon et al. (2023) and Haramati et al. (2024) found that object-centric representations are mostly beneficial for tasks requiring relational reasoning with object interactions. Additionally, Haramati et al. (2024) also

demonstrated that their agent can generalize compositionally to more objects than seen during training, both empirically and theoretically. Kim et al. (2024) provided some evidence that a slot-based State-Space Model improves compositional generalization, though the specific design elements driving this improvement remain unclear. Furthermore, Montero et al. (2024) show that a simple object-centric model reconstructs novel objects with held-out ranges of properties (e.g., color or rotation) for a single object better than a non-object-centric alternative when the models have been trained on all combinations for the rest of the objects. Rubinstein et al. (2025) and Baldassarre et al. (2025) both advocate for revisiting the original goals of object-centric learning and a departure from the evaluation of these representations solely or mostly on (unsupervised) image segmentation. Concretely, their downstream tasks consist of out-of-distribution (OOD) image classification or scene classification and action recognition in videos, respectively.

## 3. Problem Setup

This section describes how the visual datasets are generated to systematically vary compositional generalization difficulty (§3.1), the vision encoders and downstream VQA models that are compared (§3.2), and the scope and limitations of this setup (§3.3).

### 3.1. Dataset Generation

Inspired by Kim et al. (2024), we create datasets in the style of CLEVRTex (Karazija et al., 2021), Super-CLEVR (Li et al., 2023), and MOVi-C (Greff et al., 2022). For each base dataset, we create training splits with progressively smaller subsets of all possible objects, resulting in increasingly harder COOD generalization problems. For example, for CLEVRTex, each object is defined by a triplet of properties (shape, size, and material) yielding 192 unique objects (see Appendix A for details about the other datasets). We then randomly select 3–6 objects from the set of allowed objects per scene. We render images using Blender[1]. As a result, we obtain three training datasets per base dataset—CLEVRTex, Super-CLEVR, and MOVi-C easy, medium, and hard—each time decreasing the diversity by roughly halving the number of admissible objects. Every training set consists of 48k images: 40k for training, 4k for validation, and 4k for in-distribution testing. Finally, we generate a COOD test set for each base dataset, each containing 4k images using the remaining 20% of objects. As a robustness check, we also evaluate MOVi-C with unseen (OOD) backgrounds (Table 9). The performance is very similar to the main setting, suggesting that the main challenge comes from generalizing to novel objects rather than novel backgrounds.

[1] https://www.blender.org/

Our goal is to evaluate the quality of representations using VQA. Thus, for each image, we generate multiple question–answer pairs, using the generation approach of Johnson et al. (2017) adapted to CLEVRTex and MOVi-C, and the existing implementation for Super-CLEVR (Li et al., 2023) (see Appendix A). This produces 42 question–answer pairs per image on average, resulting in roughly 1.7M per training set and 170k per test set (ID & COOD).

**Why VQA?** We choose VQA as the main evaluation framework because it simultaneously tests multi-object handling and spatial as well as property-level relations within a single unified evaluation. Single-attribute classification, e.g., classifying the shape of an object, probes attribute binding in isolation only and does not include reasoning over novel property combinations. VQA further covers a range of question types—counting, attribute queries, existence, and relational comparisons—and remains applicable to any vision representation in our setup, unlike generative protocols such as Kim et al. (2024).

### 3.2. Models and Evaluation

**Setup.** To evaluate VQA, we follow the setup of Mamaghan et al. (2025). The downstream VQA model is a transformer that receives concatenated text and image representations as input and outputs a class label (details in Appendix B.3). We report results for two different sizes: a small 2-layer variant (TF 2) and a larger 5-layer variant (TF 5). We also ablate downstream capacity: larger variants do not consistently improve COOD performance (Table 11). This supports using TF 2 and TF 5 as our main downstream models. Questions are encoded by a pretrained T5-base model (Raffel et al., 2020). The answers are represented as 28 (CLEVRTex), 106 (Super-CLEVR), or 48 (MOVi-C) distinct labels, which include "yes", "no", natural numbers up to the maximum number of objects, and all possible values of object properties (including part names for Super-CLEVR).

To gauge dataset difficulty, we train two additional baselines: a naive question-only baseline using only the questions as inputs and a ground-truth oracle that supplies the true object properties of all visible objects in the scene as "image representations" for the downstream model. After training, each downstream model is evaluated on its corresponding in-distribution (ID) and compositional out-of-distribution (COOD) test set at every training checkpoint.

**Vision Models.** First, we evaluate the dense representations of two strong pretrained vision models: DINOv2 ViT-S/14 (Oquab et al., 2024) and SigLIP2 ViT-B/16 (Tschannen et al., 2025). We then consider different approaches of transforming these representations to study how COOD performance is affected. In particular, we pretrain an *object-centric model* for every dataset variant by reconstructing the pretrained dense representation with a Slot Attention

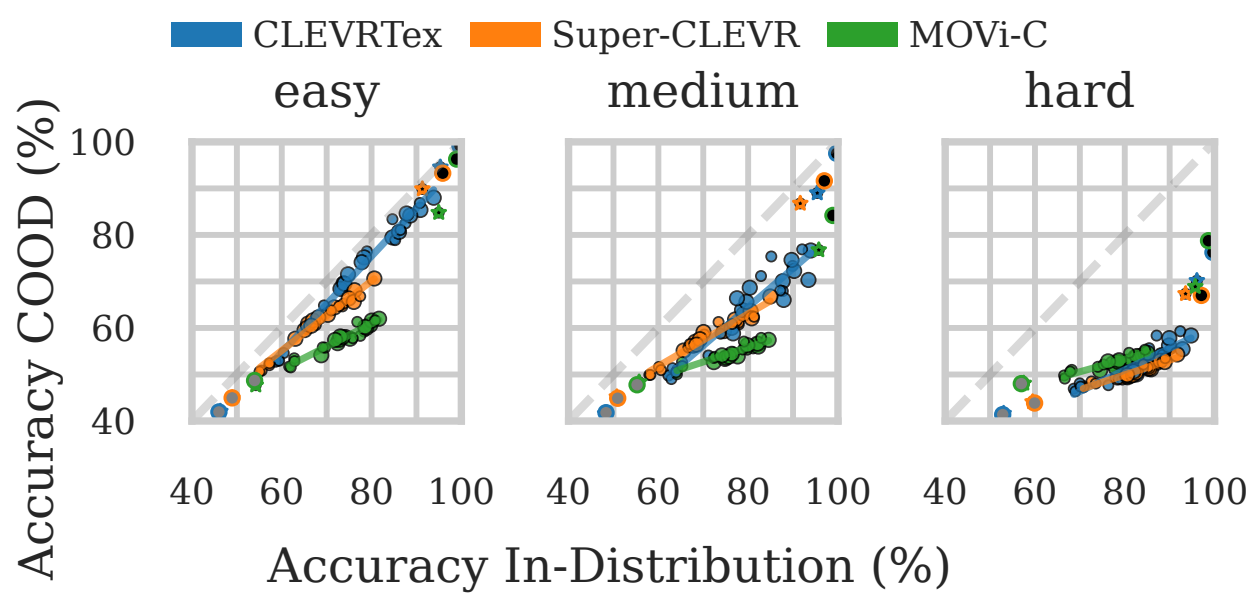

*Figure 2.* **ID and COOD VQA accuracies are strongly correlated (Pearson and Spearman $> 0.9$; $p < 0.01$).** End-of-training results for CLEVRTex, Super-CLEVR, and MOVi-C (easy, medium, hard) across all image representations (small or large point) and downstream models. Oracle is in the top-right (black), and the question-only baseline is in the bottom-left (gray).

(Locatello et al., 2020) bottleneck (Seitzer et al., 2022). This yields DINOSAURv2 (Didolkar et al., 2024) and, to the best of our knowledge, the first object-centric SigLIP2 variant, *SigLIPSAUR2*. Architectural and hyperparameter details are in Appendix B.1. As an alternative to Slot Attention (SA), we also run k-means on the set of dense patch tokens to extract a set of cluster centroids representing the image (as Baldassarre et al. (2025)).

The original vision encoders and their respective OC counterparts produce image representations of different sizes, which strongly impacts the downstream model's FLOPs (see Appendix C for details). Concretely, in our setting, the number of tokens and token dimensions are: $[256, 384]$ for DINOv2 vs. $[7, 256]$ for DINOSAURv2, and $[196, 768]$ for SigLIP2 vs. $[7, 256]$ for SigLIPSAUR2. We follow common practice and use 7 slots (max objects + background) by default. Ablating this choice, slightly more slots can improve COOD (Table 10). To enable a fair comparison, we change the size of the image representation with downstream model variants that include a single cross-attention layer immediately after the vision encoder output. This layer modifies the size of the image representation by using a number of learned queries matching the target size. We evaluate both increasing the size of the OC representation to the size of the original vision encoder, or, vice versa, decreasing the representation size of the original vision encoder. The latter could be seen as a possible alternative to Slot Attention. The cross-attention layer is trained jointly with the downstream model, and results in identical compute requirements for these variants.[2] We also experimented with replacing the Slot Attention bottleneck, e.g., in DINOSAURv2, with a cross-attention module, and replacing the downstream model's cross-attention layer with Slot Attention. Both attempts yielded suboptimal results, suggesting either that more extensive tuning is needed or that these substitutions are ill-suited for the VQA task.[3]

[2]We choose to ignore the compute from the cross-attention layer as the goal is to compare the performance of different representations fairly with respect to image representation size.

### 3.3. Limitations

While our benchmark is designed to be controlled and visually rich, several limitations remain. First, all three visual worlds are synthetic. This is essential for our setup, since the benchmark requires precise control over which object-property combinations exist and which appear during training, which is infeasible for natural images where attribute co-occurrence is governed by dataset biases. Extending the evaluation to real images is therefore a valuable but non-trivial direction for future work. Second, we focus on a single notion of compositionality: object-property composition. Third, all OC models share the same pretraining objective: reconstructing dense features (Seitzer et al., 2022; Mamaghan et al., 2025; Didolkar et al., 2024), a popular and competitive approach. Alternative objectives (e.g., pixel-space reconstruction, or contrastive OC objectives) are left for future work. Finally, our analysis centers on Slot Attention (Locatello et al., 2020) as the OC mechanism with an MLP decoder, while architecturally distinct OC families (e.g., diffusion- or transformer-decoder-based) lie outside the scope of this study.

## 4. Experiments

**Summary.** Across CLEVRTex, Super-CLEVR, and MOVi-C, we study how training diversity, downstream compute, and sample size affect in-distribution (ID) and compositional out-of-distribution (COOD) VQA performance. Reducing training diversity makes the ID task easier but COOD generalization harder (§4.1; Fig. 2). Comparing representation types, object-centric representations are most beneficial in harder COOD regimes and with smaller downstream models, while dense representations can match or surpass on easier regimes with larger compute budgets (§4.2; Tables 1 and 16). Accounting for downstream FLOPs, object-centric models typically yield higher COOD accuracy at constrained compute budgets (§4.3; Fig. 3). Finally, varying sample size shows that object-centric representations are more sample efficient, whereas dense encoders match or surpass only with sufficiently large and diverse data and larger downstream models (§4.4; Figs. 4 and 5).

For easier readability, we often refer to a single base dataset in the following and explicitly mention if trends differ across datasets. All results can be found in Appendix D.

[3]For a discussion of cross-attention as an alternative to Slot Attention, and why it may be suboptimal for this kind of pretraining, see Wu et al. (2023).

*Table 1.* **Object-centric representations are better for harder generalization, especially with the smaller downstream model.** VQA accuracy (%) of both downstream models (TF 2 & 5) on the respective compositional generalization test sets for all DINOv2-based models trained on easy (E), medium (M), and hard (H) training sets. We compute deltas relative to the original pretrained vision encoder and list the number of tokens in the representations ("size"). With TF 5, DINOv2 is better on easy, while DINOSAURv2 matches or slightly surpasses it on hard.

| | | | CLEVRTex | | | Super-CLEVR | | | MOVi-C | | |
|---|---|---|---|---|---|---|---|---|---|---|---|
| | Model | Size | E | M | H | E | M | H | E | M | H |
| TF 2 | DINOv2 | 256 | 69.5 | 58.8 | 50.0 | 60.9 | 57.0 | 49.7 | 57.5 | 53.6 | 51.4 |
| TF 2 | DINOv2 + CA | 7 | -1.2 | -4.8 | -2.3 | -1.2 | -1.2 | -0.8 | -1.7 | -0.1 | +0.2 |
| TF 2 | DINOv2 + k-means | 7 | -16.5 | -9.1 | -3.1 | -10.1 | -7.1 | -2.2 | -5.8 | -2.6 | -1.6 |
| TF 2 | DINOv2 + k-means | 128 | -1.1 | +1.2 | -0.6 | -1.4 | -0.8 | -0.1 | -0.7 | -0.3 | +0.9 |
| TF 2 | DINOSAURv2 | 7 | +7.0 | +12.3 | +5.6 | -0.3 | +1.6 | +1.2 | +1.0 | +1.1 | +1.6 |
| TF 2 | DINOSAURv2 + CA | 256 | +0.1 | +9.8 | +1.0 | -3.3 | -1.9 | -0.4 | -3.3 | -1.0 | -0.6 |
| TF 5 | DINOv2 | 256 | 85.4 | 70.3 | 55.4 | 68.1 | 63.0 | 51.7 | 60.0 | 56.0 | 54.0 |
| TF 5 | DINOv2 + CA | 7 | -6.5 | -2.4 | -1.7 | -2.9 | -1.9 | -0.9 | -1.7 | -0.7 | -0.8 |
| TF 5 | DINOv2 + k-means | 7 | -32.6 | -21.3 | -9.1 | -17.5 | -13.1 | -4.5 | -8.5 | -6.0 | -4.9 |
| TF 5 | DINOv2 + k-means | 128 | -4.7 | -4.3 | -1.3 | -3.7 | -1.2 | -0.4 | +0.6 | +0.5 | -0.1 |
| TF 5 | DINOSAURv2 | 7 | -2.9 | +3.0 | +0.1 | -3.5 | -2.2 | +0.4 | -0.8 | -0.4 | 0.0 |
| TF 5 | DINOSAURv2 + CA | 256 | -5.9 | -0.3 | -1.4 | -5.3 | -3.7 | -0.6 | -2.3 | -2.1 | -2.0 |

### 4.1. The Effect of Data Diversity

**Finding I (Training Diversity)**

Training diversity controls generalization difficulty: as diversity decreases, ID accuracy increases but COOD accuracy drops, widening the ID–COOD gap across datasets and encoders.

**Oracle.** We first validate that our experimental setup, including datasets and downstream models, is suitable for testing compositional generalization by establishing that a model with the "right" representation is able to solve the task in-distribution (ID) but might still struggle in compositional out-of-distribution (COOD) generalization as the difficulty increases. Specifically, we train an oracle that uses the ground-truth object properties as image representation. For all training datasets, the oracle can achieve nearly perfect ID test accuracy (Fig. 2), given a sufficiently large downstream model. However, its compositional generalization drops notably when trained on smaller subsets of the full visual space. As an example, it still reaches almost 100% on CLEVRTex COOD by training on easy, but struggles to even reach 80% when training on hard.

**ID vs. COOD.** Having established the suitability of our setup, we now investigate models with learned image representations. Evaluating all vision encoders, as depicted in Fig. 2, we observe a consistent and intuitive pattern: as we constrain the diversity of the training data—thereby increasing generalization difficulty—the models' in-distribution accuracy improves due to fewer visual combinations to learn. However, this simplification in ID tasks simultaneously intensifies COOD challenges, as models must generalize from fewer learned combinations to the fixed COOD test set. This is consistent across all base datasets and vision encoders.

### 4.2. The Effect of Image Representation Type

**Finding II (Representation Type)**

Representation type matters: object-centric representations are better for harder generalization, especially with smaller downstream models, while dense representations can match or surpass them in easier settings. When matching representation size, cross-attention resizing or k-means generally fall short of Slot Attention.

**OC vs. Dense.** We begin by comparing the OC representations to their dense counterparts, ignoring the differences in their representation sizes. The OC versions are almost always better at compositional generalization with the smaller downstream model (Table 1). Concretely, the changes in generalization for DINOSAURv2 relative to DINOv2 on TF 2 range from -0.3% to +12.3% (absolute) across all training datasets, and are especially large on CLEVRTex. The trend is consistent across vision encoder families for SigLIP2-based representations (Table 16). When increasing the power of the downstream model (Table 1: TF 5), the dense representations are better for easier generalizations (easy) but lose their benefit when the OC representations either match or slightly surpass them for harder generalizations (hard). For example, the differences in hard generalization with the larger downstream model for DINOSAURv2 and its dense counterpart are from 0.0% to +0.4%. This is

*Table 2.* **Property prediction on CLEVRTex (COOD).** Mean±std over three seeds of the downstream MLP. Higher is better for accuracy (Material, Shape, Size); lower is better for position MSE (X, Y). DINOSAURv2 outperforms DINOv2 on the majority of cases, with the gap widening on harder generalizations, particularly for material and shape, while position MSE is consistently and substantially lower for DINOSAURv2 across all difficulties.

| Diff. | Model | Material↑ | Shape↑ | Size↑ | X↓ | Y↓ |
|---|---|---|---|---|---|---|
| E | DINOv2 | 88.3±1.9 | **81.4±6.8** | **91.3±3.0** | .019±.005 | .004±.001 |
| | DINOSAURv2 | **88.8±0.0** | 78.0±0.2 | 88.7±0.6 | **.004±.000** | **.001±.000** |
| M | DINOv2 | 63.4±3.9 | 64.3±1.9 | 83.2±2.2 | .044±.002 | .007±.000 |
| | DINOSAURv2 | **77.3±0.2** | **75.5±0.2** | **84.2±0.6** | **.005±.000** | **.001±.000** |
| H | DINOv2 | 47.3±6.7 | 49.3±1.3 | **70.7±3.4** | .061±.004 | .010±.000 |
| | DINOSAURv2 | **60.7±0.6** | **61.8±1.6** | 67.5±1.4 | **.009±.000** | **.002±.000** |

*Table 3.* **Trends are stable across seeds (COOD).** Mean±std over three seeds of the downstream model on CLEVRTex easy (E), medium (M), and hard (H). With TF 2, DINOSAURv2 outperforms DINOv2 across all difficulties; with TF 5, the dense encoder surpasses DINOSAURv2 only on the easy setting, while the OC advantage persists on the medium and hard generalizations.

| | Model | E | M | H |
|---|---|---|---|---|
| **TF 2** | DINOv2 | 69.9±0.3 | 59.5±0.9 | 49.6±0.3 |
| | DINOSAURv2 | **76.3±0.9** | **71.3±0.1** | **55.1±0.7** |
| **TF 5** | DINOv2 | **84.2±1.4** | 70.5±1.3 | 54.8±0.7 |
| | DINOSAURv2 | 82.9±0.6 | **72.9±0.4** | **55.2±0.3** |

again consistent for SigLIP2-based models (Table 16).

**K-means vs. Slot Attention.** Comparing OC-like representations in Table 1, i.e., the pretrained Slot Attention and k-means variants, we observe that the k-means representations with the same number of tokens as the SA versions (7 tokens) are quite worse in compositional generalization. We hypothesize that this is due to the ineffectiveness of k-means, a method that does not use additional training, in drastically reducing the number of visual tokens (e.g., from 256 to 7 for DINOv2) by simply taking the centers of each cluster, compared to a "soft k-means" as performed by Slot Attention (Locatello et al., 2020). This is in contrast to (Baldassarre et al., 2025), where a small number of tokens was often sufficient. This discrepancy may be explained by VQA being a task that requires more fine-grained visual information compared to the more global or coarse-grained tasks in (Baldassarre et al., 2025). In order to partially overcome this, we increase the number of clusters used for k-means, i.e., the number of visual tokens here, to 128 such that there is still a reduction from the original representation while getting the best performance compared to using any number of fewer tokens (for details see Appendix B.2). The improved k-means representation, at the cost of using more tokens, is sometimes able to match the generalization capabilities of the SA versions with the bigger downstream model, especially on MOVi-C, while still falling behind for the smaller one (Table 1).

**Reduction with Cross-Attention.** To match the capacity of OC models, we decrease dense image representations with a single cross-attention (CA) layer. This reduction is generally less effective than Slot Attention: with the smaller downstream model, DINOv2+CA underperforms DINOSAURv2, and SigLIP2+CA underperforms SigLIPSAUR2 on COOD (Tables 1 and 16). With the larger downstream model, the gap narrows but does not disappear. We hypothesize that Slot Attention's structured bottleneck makes object-level information easier to extract, especially with limited-capacity downstream models (Mamaghan et al., 2025).

**Expansion with Cross-Attention.** Conversely, expanding an OC representation with CA is rarely helpful. Across datasets and downstream model sizes, DINOSAURv2+CA and SigLIPSAUR2+CA typically underperform both their original OC representations and the dense encoders at the matched (larger) representation size (Tables 1 and 16).

**Robustness across seeds.** To verify that the observed trends for dense and OC representations are not driven by seed variability, we repeat the CLEVRTex DINOv2 vs. DINOSAURv2 comparison across three seeds (Table 3). Again, the dense representations only generalize better than their OC counterparts with the larger downstream model (TF 5) on the easy generalization (E).

**Beyond VQA: property prediction.** To check whether the observations extend beyond VQA, we train a one-hidden-layer MLP on top of the frozen DINOv2 and DINOSAURv2 representations to predict per-object material, shape, size, and 2D position, following Seitzer et al. (2022). Results across three seeds on CLEVRTex are reported in Table 2. The benefit of DINOSAURv2 becomes most apparent on the harder generalizations (medium, hard), where the difference is especially large for material and shape classification. For position prediction, DINOSAURv2 reaches consistently lower MSE than DINOv2 across all difficulties, whereas DINOv2 representations are generally better or competitive for Size prediction.

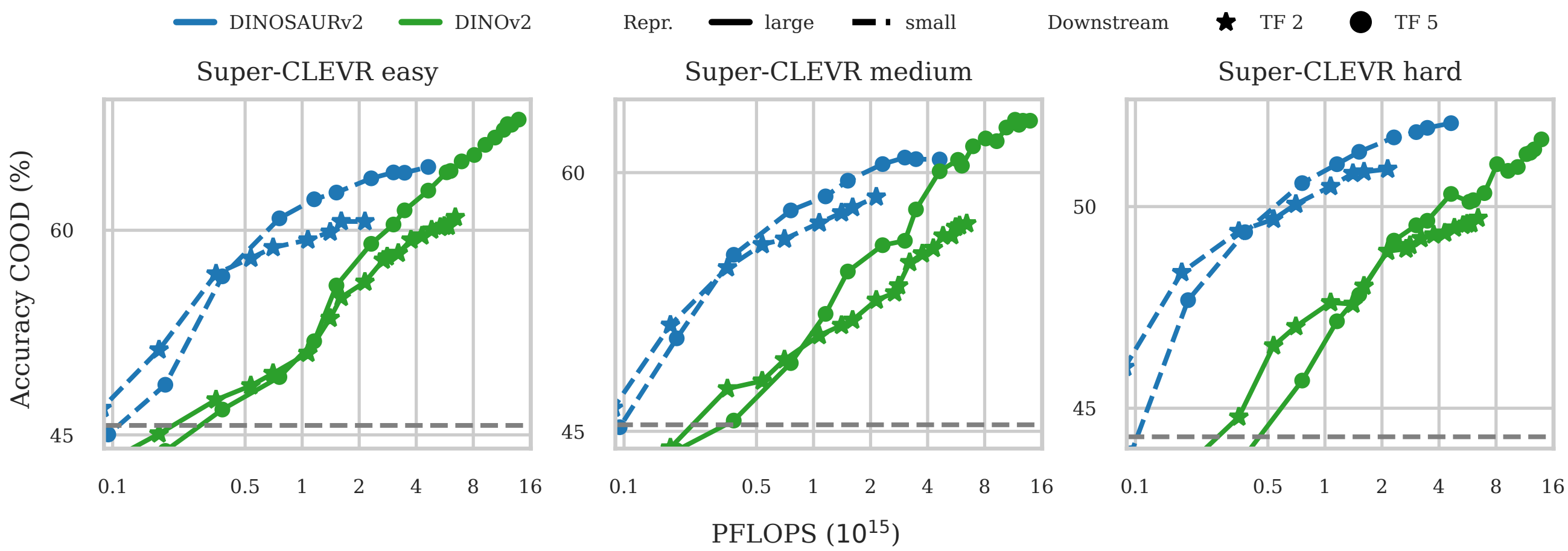

*Figure 3.* **Object-centric representations are more compute-efficient.** COOD VQA accuracy on Super-CLEVR easy (left), medium (middle), and hard (right) versus downstream compute (log FLOPs) for DINOv2-based models; the question-only baseline is dashed gray. Dense representations slightly outperform on easy but do not match OC performance on hard even with 3 $\times$ compute.

### 4.3. The Effect of Downstream Compute

**Finding III (Compute)**

At constrained downstream compute, object-centric representations achieve higher COOD. Dense representations can match or surpass them on easier settings, but only with substantially more compute.

**Small Compute Budgets.** The COOD accuracies at different compute budgets and training difficulties of Super-CLEVR for the DINO-family are shown in Fig. 3. For small compute budgets—up to roughly four PFLOPs, the end of training for smaller image representations—and generalization difficulties, Slot Attention-based OC representations consistently outperform all other representations.

**Easy Generalizations.** To surpass the COOD performance of DINOSAURv2 for easier generalization tasks, the non-OC counterpart, DINOv2, requires substantially more downstream compute. Even then, the eventual final accuracy gain is modest ($\leq$ 3.5%). For Super-CLEVR easy in Fig. 3 (left), DINOv2 reaches the best generalization accuracy of DINOSAURv2 with 1.5 $\times$ the compute and improves $+3.5\%$ at the end after consuming 3 $\times$ the computational resources. The same observations hold for the SigLIP2-based models (Table 16).

**Hard Generalizations.** For the settings with the hardest compositional generalization, for example, Super-CLEVR hard (Fig. 3 right), small object-centric representations consistently match or outperform their non-object-centric counterparts within the same backbone family at any given compute budget. Even when granting original vision encoders up to 3 $\times$ the compute, they often fail to surpass the object-centric representations (Table 16).

**Small Downstream Model.** Considering the performance of representations across downstream models, employing a smaller downstream model for the best COOD performance is justified only under very constrained compute budgets (for example, below 0.5 PFLOPs in Fig. 3). Under these limited compute conditions, DINOSAURv2's or SigLIPSAUR2's small image representation paired with the small downstream model consistently outperforms all alternatives.

### 4.4. The Effect of Sample Size

**Finding IV (Sample Size)**

Object-centric representations are more sample-efficient: they achieve stronger COOD with fewer images and smaller downstream models, whereas dense representations surpass them only with sufficiently large and diverse datasets and larger downstream models.

**Varying Sample Size.** To examine how the amount of training data and its diversity affect compositional generalization, we train both the original vision encoders and their object-centric counterparts on subsets of all datasets. Specifically, we vary the sample size, defined as the number of training images, from $2^{10}$ (1024) up to $2^{15}$ (32768), each paired with corresponding question–answer pairs, and compare results to training on the full set of 40k images.

**Sample Size vs Highest Diversity.** When training on subsets of the dataset with the highest diversity, here shown for MOVi-C easy in Fig. 4, the object-centric models consistently achieve better compositional generalization than their non-object-centric counterparts across all sample sizes when paired with the small downstream model (Fig. 4 left). In

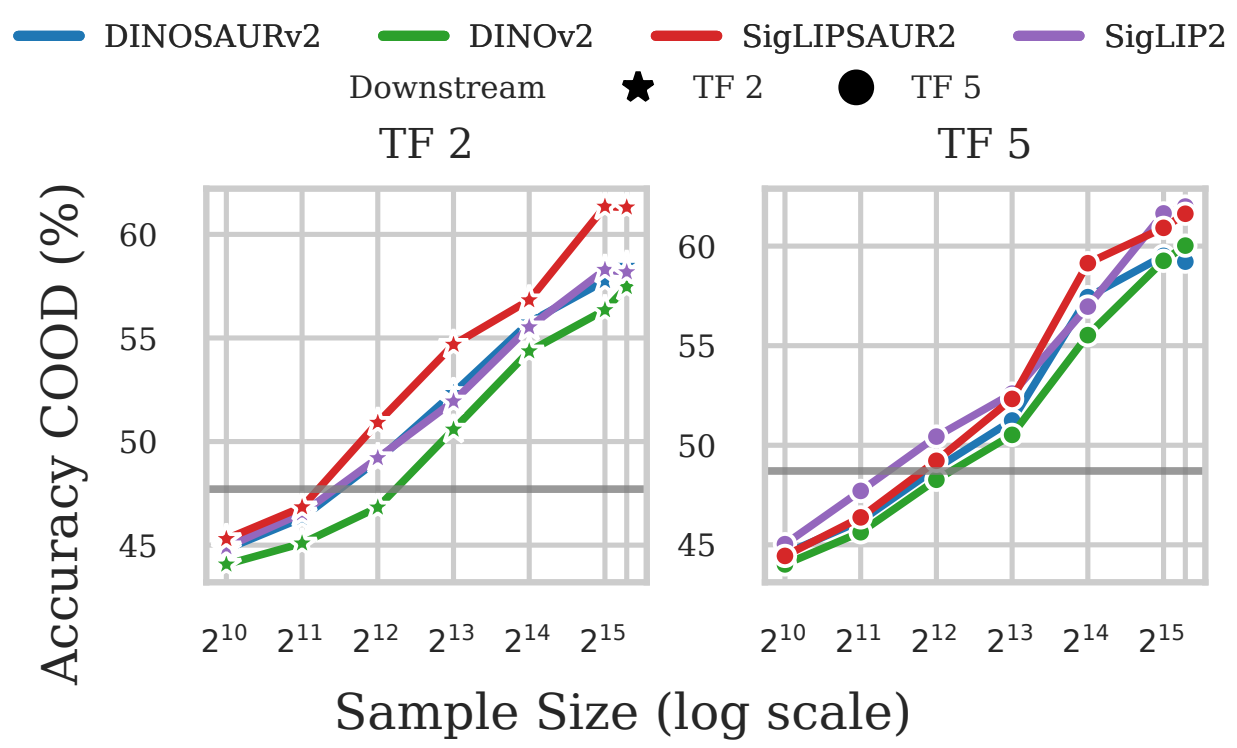

*Figure 4.* **Object-centric representations are more sample-efficient.** COOD VQA accuracy on MOVi-C easy versus training set size for TF 2 (left) and TF 5 (right). The question-only baseline trained on the full dataset is shown in gray. The OC advantage is strongest at smaller sample sizes and with TF 2. Dense representations only catch up or slightly surpass them at the full 40k samples with TF 5.

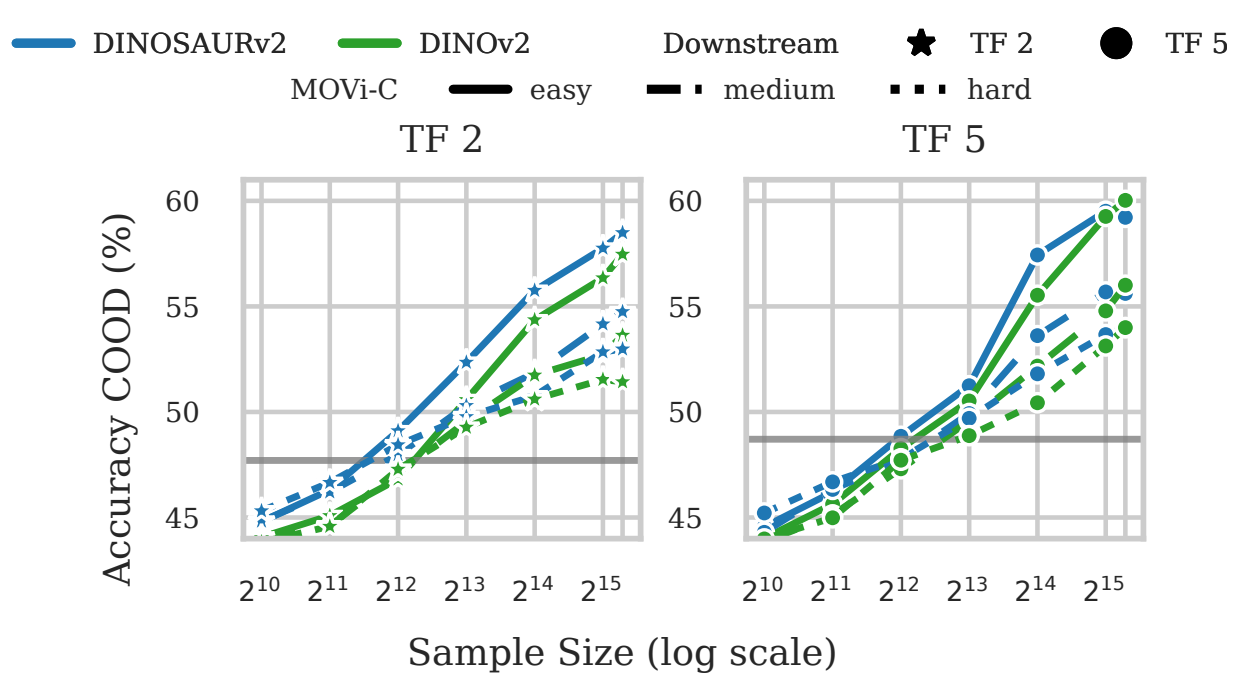

*Figure 5.* **OC models generalize better at low training diversity.** COOD VQA accuracy for DINOv2 and DINOSAURv2 trained on MOVi-C subsets across sample sizes and diversities (easy–hard) for TF 2 (left) and TF 5 (right). The question-only baseline (full data) is shown in gray. Dense DINOv2 only overtakes DINOSAURv2 at the largest sample size for easier generalizations. Under lower diversity or fewer data points, DINOSAURv2 generalizes better or as well.

contrast, with more computational resources (larger downstream model, right), non-object-centric representations can match or slightly surpass object-centric representations at larger sample sizes, here only at the largest sample size of the full dataset (40k). This indicates that object-centric models are more sample efficient, likely because their smaller representations explicitly decompose the visual content of objects into different tokens, i.e., slots.

**Sample Size vs Diversity.** Across sample sizes, models trained on more diverse data—i.e., seeing more unique objects during training (easy—hard)—almost always generalize better compositionally, especially at larger sample sizes. For MOVi-C in Fig. 5, higher diversity consistently yields better generalization for both DINOv2 and DINOSAURv2. For all models and datasets, there exists a breakpoint where doubling the number of samples improves compositional generalization more for higher diversities than for lower diversities. For MOVi-C in Fig. 5, both models struggle to clear the question-only baseline (trained on the full data) at low sample sizes, and performance is similar across diversities up to $2^{13}$ (8k) samples. Doubling from $2^{13}$ to $2^{14}$ yields larger gains for higher diversities for both models, with the strongest improvement for DINOSAURv2. The location of this breakpoint can depend on the dataset, model, and diversity, and performance at lower diversity can plateau or even decrease at higher sample sizes. Lastly, nearly all models match or surpass their compositional performance on the second-most diverse dataset (medium, full sample) with only $2^{14}$ images (around 40% of the full data) from the most diverse dataset (easy), indicating that diversity is more important than sample size for generalization.

**OC vs Dense.** Focusing now on the difference between models, the object-centric representation is better at generalizing than its dense counterpart for all diversities across all sample sizes for the small downstream model, sometimes even comparing across diversities (e.g. for CLEVRTex with TF 2 see Figs. 16 and 17). For the bigger downstream model, the dense representations only surpass the OC model at higher diversities and sample sizes. For MOVi-C in Fig. 5 (right), DINOv2 only surpasses DINOSAURv2 on easy and medium for the highest sample size. If we restrict one or both of diversity and sample size enough, the object-centric representation generalizes equally well or better.

## 5. Conclusion

In this work, we systematically evaluated the compositional generalization capabilities of object-centric representations in fully controlled and visually rich settings. Using a VQA benchmark across CLEVRTex, Super-CLEVR, and MOVi-C, we find that object-centric models (DINOSAURv2, SigLIPSAUR2) are superior on harder compositional generalization settings and generally preferable when diversity, sample size, or downstream compute is constrained. Dense encoders (DINOv2, SigLIP2) can catch up and sometimes surpass object-centric representations on easier settings, but typically only with sufficiently large and diverse training data and a larger downstream model, which requires substantially more downstream compute.

These findings reinforce the potential of object-centric approaches for tasks requiring systematic compositional reasoning and highlight the need for further exploration into their applications beyond synthetic benchmarks. Future work may extend this by investigating the effectiveness of object-centric learning in real-world scenarios, incorporating more diverse datasets, and optimizing architectural choices to enhance performance.

## Acknowledgements

The authors would like to thank Max Horn for insightful discussions and support throughout this work.

This work was partially supported by the Helmholtz Foundation Model Initiative and the Helmholtz Association. The authors gratefully acknowledge the Gauss Centre for Supercomputing e.V. (www.gauss-centre.eu) for funding this project by providing computing time through the John von Neumann Institute for Computing (NIC) on the GCS Supercomputer JUPITER — JUWELS (Jülich Supercomputing Centre, 2021) at Jülich Supercomputing Centre (JSC). Furthermore, the authors appreciate the computational resources provided by the National High Performance Computing Centre (www.nhr.kit.edu). The research presented is supported by the TUM Georg Nemetschek Institute Artificial Intelligence for the Built World.

## Impact Statement

This paper presents work whose goal is to advance the field of Machine Learning. There are many potential societal consequences of our work, none which we feel must be specifically highlighted here.

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

# A. Data Generation

**Main experiments.** The used attributes for image generation are in Table 4 for CLEVRTex, Table 5 for Super-CLEVR and Table 6 for MOVi-C, with some example images with question–answer pairs in Fig. 6, Fig. 7 and Fig. 8, respectively. For each base dataset, we define an object as the combination of all its attributes, and create three training datasets, which we label as easy (containing 80% of all possible objects), medium (40%), and hard (20%). We reserve the holdout 20% of object–property combinations for testing compositional generalization. During initial investigations, we found that for Super-CLEVR the generalization problem at these proportions is not hard enough yet, i.e., generalization behaves as in-distribution. This is likely due to the many factors of Super-CLEVR, of which not all are equally important for answering questions. Therefore, for Super-CLEVR only, we reduced the proportions to 10% (easy), 5% (medium) and 1% (hard).

*Table 4.* Attributes for the image and question generation for CLEVRTex.

| **Shape** (8) | **Size** (3) | **Material** (8) |
|---|---|---|
| cube | small | green tiled |
| cylinder | medium | blue denim |
| monkey head | large | red fabric |
| icosahedron | | green forest |
| teapot | | red leather |
| sphere | | rocky gravel |
| cone | | rusty metal |
| torus | | white sandstone |

*Table 5.* Attributes for the image and question generation for Super-CLEVR.

| **Shape** (21) | **Size** (2) | **Material** (2) | **Color** (8) | **Texture** (4) |
|---|---|---|---|---|
| suv | small | rubber | gray | none |
| wagon | large | metal | red | checkered |
| minivan | | | blue | striped |
| sedan | | | green | dotted |
| truck | | | brown | |
| articulated bus | | | purple | |
| regular bus | | | cyan | |
| double bus | | | yellow | |
| school bus | | | | |
| chopper | | | | |
| dirtbike | | | | |
| scooter | | | | |
| cruiser | | | | |
| jet | | | | |
| fighter | | | | |
| biplane | | | | |
| airliner | | | | |
| road bike | | | | |
| utility bike | | | | |
| mountain bike | | | | |
| tandem bike | | | | |

# B. Models

## B.1. DINOSAURv2 and SigLIPSAUR2

The hyperparameters for DINOSAURv2 and SigLIPSAUR2 for all datasets can be found in Table 7.

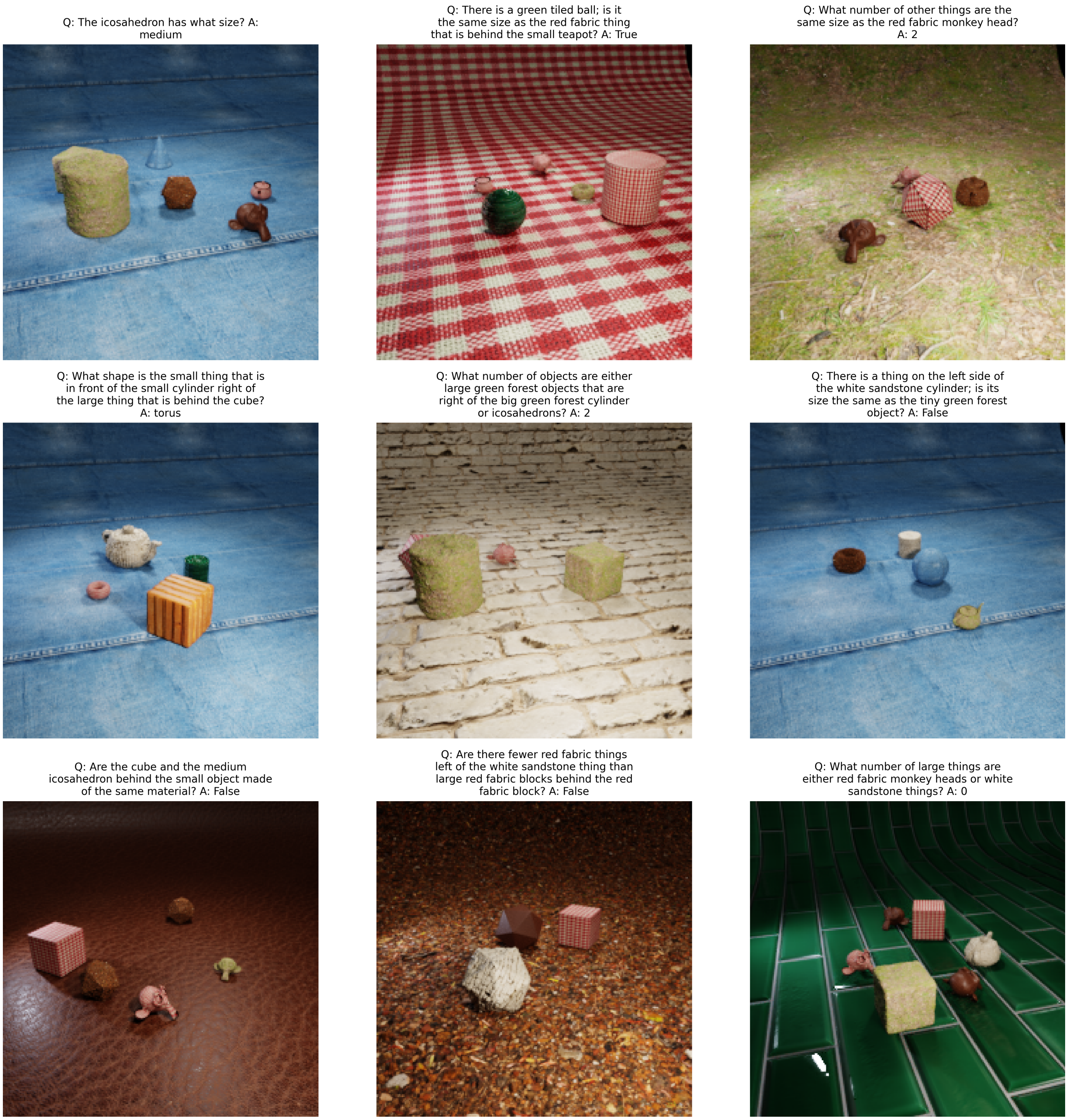

*Figure 6.* Dataset examples with question–answer pairs for CLEVRTex.

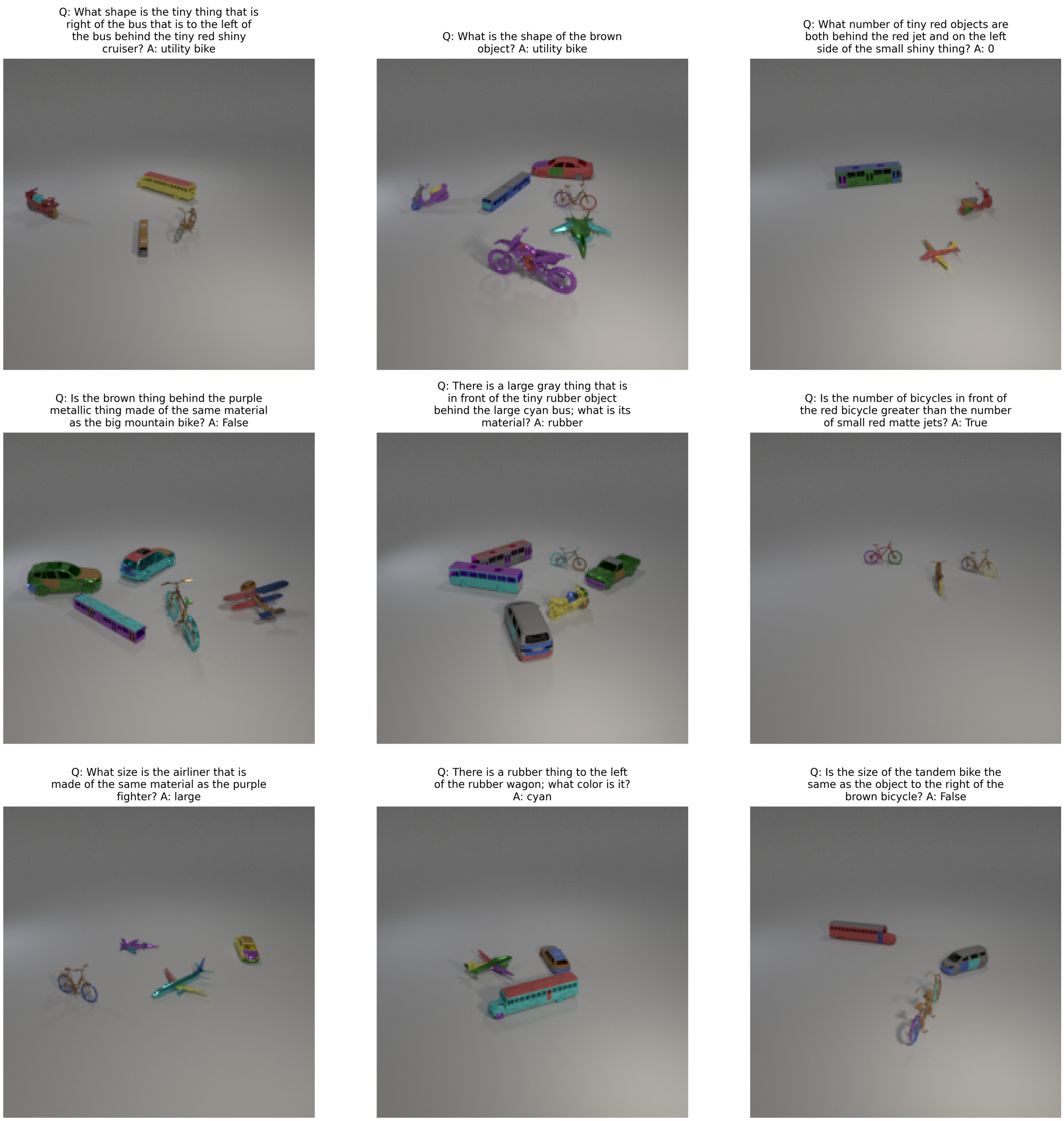

*Figure 7.* Dataset examples with question–answer pairs for Super-CLEVR.

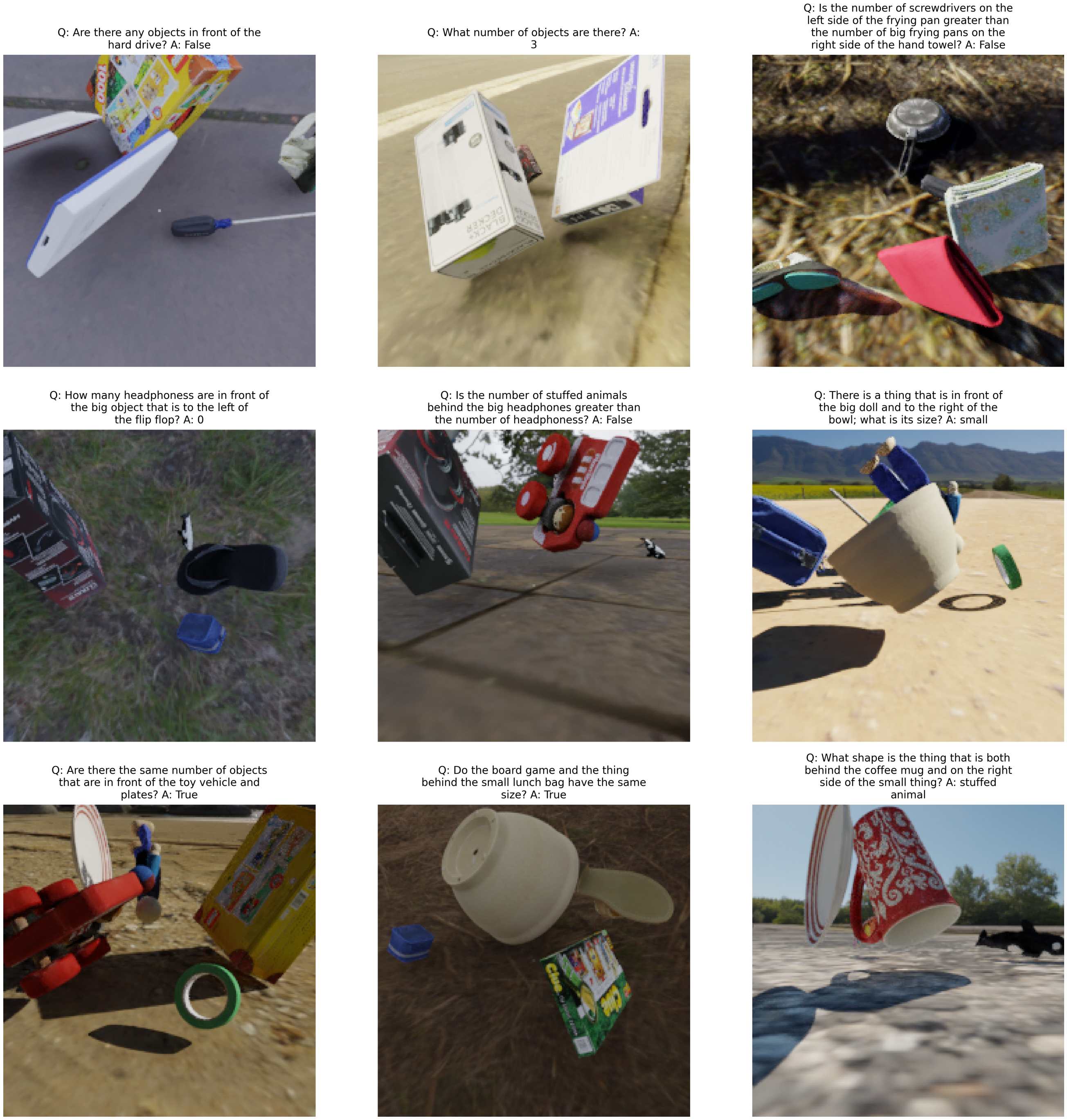

*Figure 8.* Dataset examples with question–answer pairs for MOVi-C.

### B.2. K-means

For extracting an image representation via k-means from the pretrained vision encoders, we follow Baldassarre et al. (2025). Concretely, we concatenate the global image representation $g$ (from the CLS token) with the set of centroids derived from performing k-means on the patch tokens for different numbers of clusters. After choosing the maximum number of clusters $k_{max}$, as a power of two, we use the standard sklearn `KMeans`[4] implementation for $k \in \{1, 2, \ldots, 2^{log_2(k_{max})}\}$. In contrast to Baldassarre et al. (2025), who mostly use a smaller $k_{max} = 8\,\text{or}\,16$ for their more "global" tasks, e.g., scene classification and action recognition in videos, our VQA tasks require a more fine-grained spatial understanding and knowledge of visual details to distinguish between spatial relationships (e.g., left or right) and different objects (e.g., mountain or road bike). In preliminary experiments, we tried using $k_{max} \in \{8, 16, 32, 64\}$, but image representations with $k_{max} \leq 32$ performed poorly, especially on CLEVRTex. In order to achieve reasonable performance while still resulting in a smaller image representation, therefore requiring less compute, we chose $k_{max} = 64$ which results in a representation with $128 = 1\,(\text{g}) + 1\,(\text{k=1}) + 1\,(\text{k=2}) + \ldots + 64\,(\text{k=64})$ tokens and feature dimension corresponding to the original vision encoder. For compute comparisons, refer to Appendix C.

### B.3. Downstream VQA Model

**Architecture.** We adopt a transformer-based architecture for VQA, following Mamaghan et al. (2025). We first project both image and text representations via separate linear layers (output size 126) with a dropout of 0.1, and augment them with a two-dimensional one-hot vector to indicate whether they originate from image features or text embeddings. We then add a sinusoidal positional encoding to the text embeddings. To perform classification, we use a trainable CLS $\in \mathbb{R}^{128}$ vector. We concatenate the image and text representations (plus the CLS token) and pass them through a transformer encoder with $d_{model} = 128$ and a hidden dimension of 128. The transformed CLS token is fed into a two-layer MLP (hidden dimension 128) with layer normalization, a dropout rate of 0.1, and a ReLU activation between layers. This MLP outputs a probability distribution over all possible answers.

**Training.** For all CLEVRTex, Super-CLEVR and MOVi-C variants, we train the downstream models with a batch size of 128, a learning rate of 0.0001, and a cross-entropy loss for steps defined in section Appendix C. We use downstream model variants where we vary the number of layers of the transformer encoder, either 2 or 5 layers with 64 heads. We tried changing the learning rate and schedule, including linear warm-up and/or different learning rate schedules, e.g., cosine, but all of them resulted in worse performance compared to the above setting.

## C. Compute

The base models DINOv2, SigLIP2, and DINOSAURv2/SigLIPSAUR2 produce (for image size 224) representations of shape $[256, 384]$, $[196, 768]$, and $[7, 256]$, respectively. This creates a large compute mismatch in the downstream model (Fig. 9); for example, the downstream FLOPs with the DINOv2 representation are roughly four times those with DINOSAURv2 for both transformer sizes.[5]

To control for this mismatch, we insert a single cross-attention (CA) layer with four heads immediately after the vision encoder to map between the large and small representation sizes. We also considered mapping from the input to the same output size and varying the number of CA layers or heads, but did not observe consistent improvements.

To set comparable training budgets, we train the most expensive downstream configuration (DINOv2; Fig. 9) to convergence for 600k steps, following Mamaghan et al. (2025). For each chosen checkpoint, the union of every 50k steps and a power-of-two series, we compute the corresponding number of steps for other representations using their compute ratios. Because this can yield very large step counts for smaller representations, we cap training at a fixed maximum once learning has largely saturated. The resulting step budgets are listed in Table 8.

## D. Additional Comparisons

**Main Experiments.** For completeness, we include the full main experiment results. The full ID–COOD accuracy curves across all datasets and image representations, for CLEVRTex, Super-CLEVR, and MOVi-C respectively, are shown in Figs. 10 to 12. The corresponding tables with main ID and COOD accuracies are in Tables 12 to 15 with the full COOD

[4] `https://scikit-learn.org/`
[5] `https://github.com/facebookresearch/fvcore`

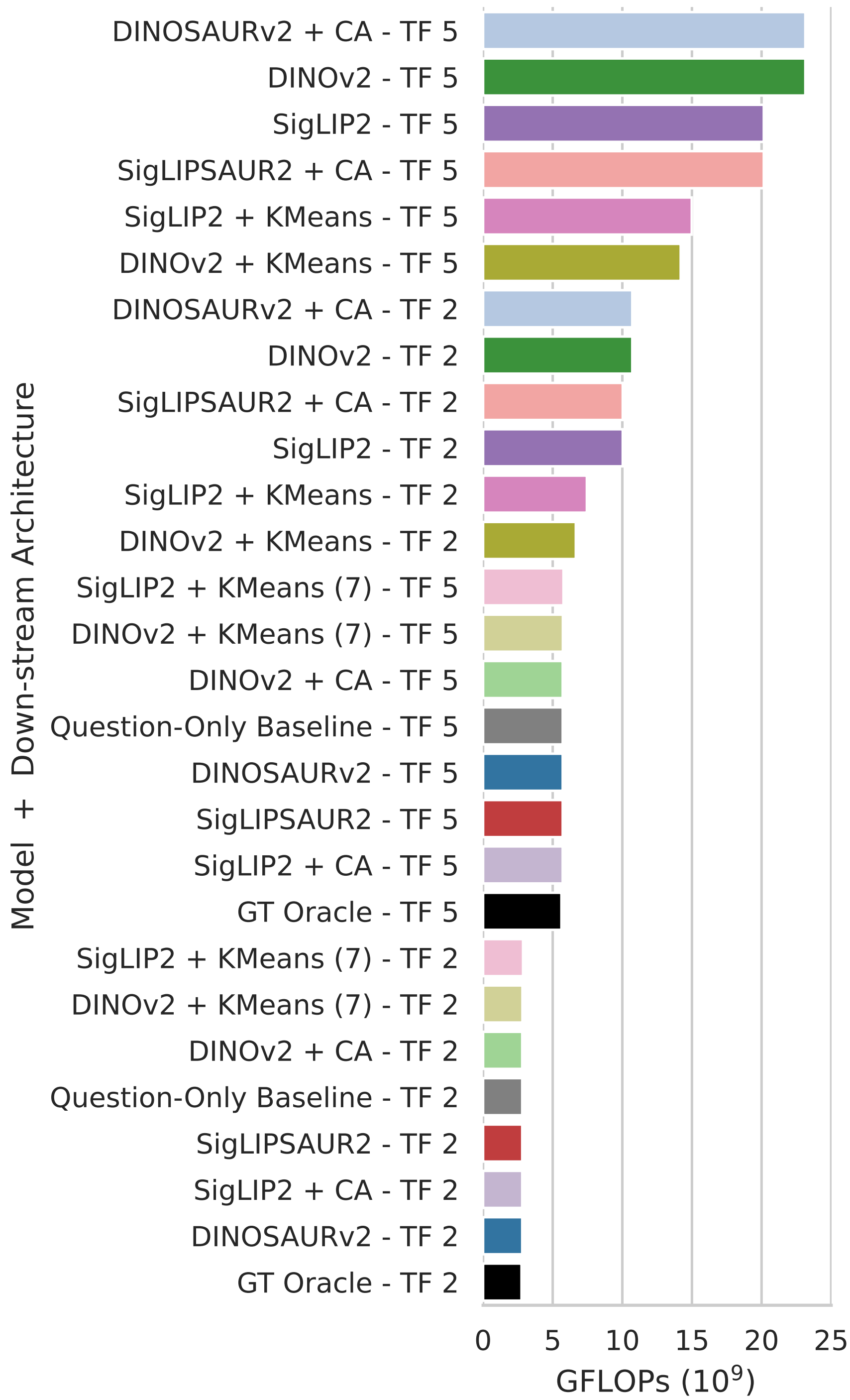

*Figure 9.* GFLOPs for one step of the downstream model for image representations with both the smaller (TF 2) and bigger transformer downstream model (TF 5), ignoring the compute needed for resizing with the cross-attention layer.

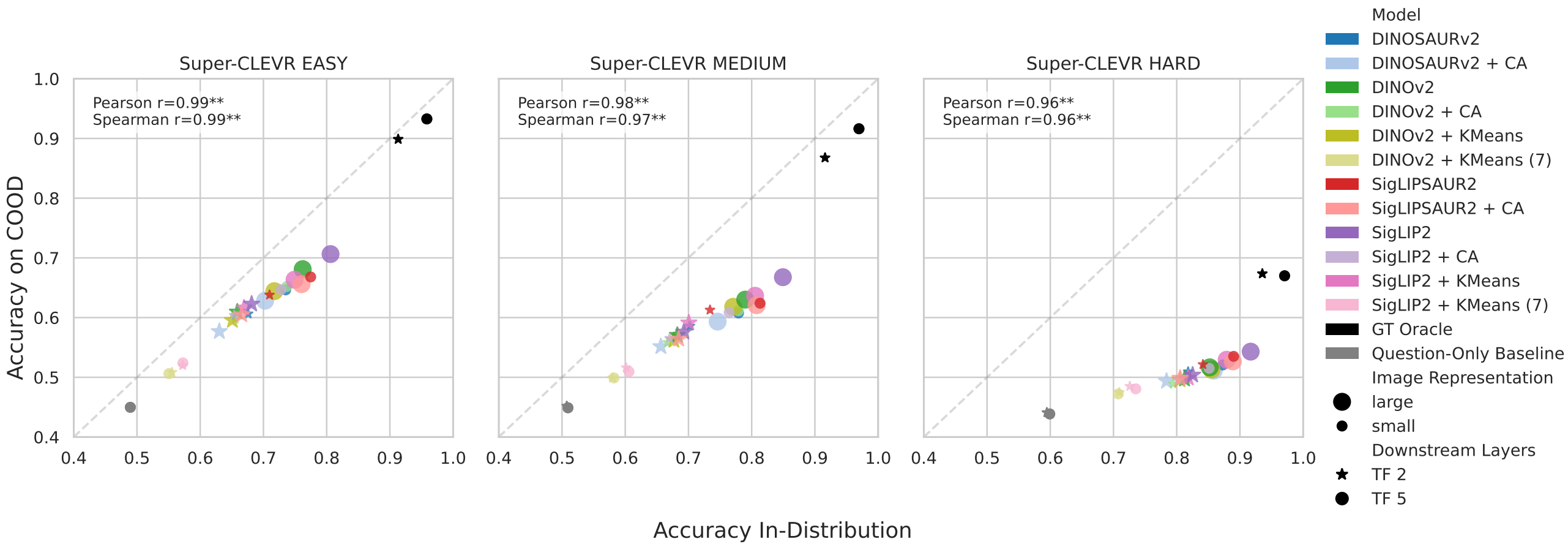

*Figure 10.* ID and COOD VQA accuracies are strongly correlated (Pearson and Spearman; $p < 0.01$). End-of-training results for Super-CLEVR (easy, medium, hard) across all image representations and downstream models. The ground-truth oracle is in the top right (black) and the question-only baseline in the bottom left (gray).

delta accuracies summarized in Table 16. Finally, full accuracy–compute trade-offs (accuracy vs. downstream FLOPs) are provided in Figs. 13 to 15.

**Sample Size.** We report full sample-efficiency results by varying the number of training images while keeping the COOD test set fixed. Results for DINOv2- and SigLIP2-based models (for both TF 2 and TF 5) across CLEVRTex, Super-CLEVR, and MOVi-C are shown in Figs. 16 and 17.

**OOD Background.** As an example of real-world noise, we consider unseen (OOD) backgrounds and evaluate on a MOVi-C COOD test set with unseen backgrounds (Table 9). We make two observations. First, all previous trends remain: in particular, OC representations continue to perform better on the harder compositional tasks. Second, the accuracies are overall very similar to those on the MOVi-C COOD test set without OOD backgrounds (compare Tables 13 and 15), indicating that the main difficulty of the task stems from novel objects rather than novel backgrounds.

**Choice of number of slots.** For our controlled benchmarks, we followed the common practice for object-centric representations to set the number of slots to the maximum number of objects per image plus one background slot (Seitzer et al., 2022; Mamaghan et al., 2025; Locatello et al., 2020), which motivates the choice of 7 slots. We additionally ablate the number of slots and find that 7 slots are often not optimal for compositional generalization (Table 10), and using slightly more slots (9 or 11) typically performs better. This further strengthens our main findings (Table 1), as the additional compute from a few extra slots, i.e., visual tokens, is negligible compared to the larger representation of the dense counterpart (DINOv2: 256 tokens), and OC representations further close or increase the performance gap.

**Ablations.** To assess the effect of downstream model capacity, we conducted additional ablations varying the number of layers (TF 10 and TF 15), the hidden dimension (TF 5 with $d_{hidden} = 256$ or 512), and both model and hidden dimension (TF 5 with $d_{model} = 256$ and $d_{hidden} = 1024$). As summarized in Table 11, none of these alternatives consistently outperform the default TF 5 downstream model (most are worse) despite using more compute. Moreover, the downstream model is sufficiently expressive to achieve 100% in-distribution accuracy when given the *correct* image representation, i.e., ground-truth object properties (Fig. 2).

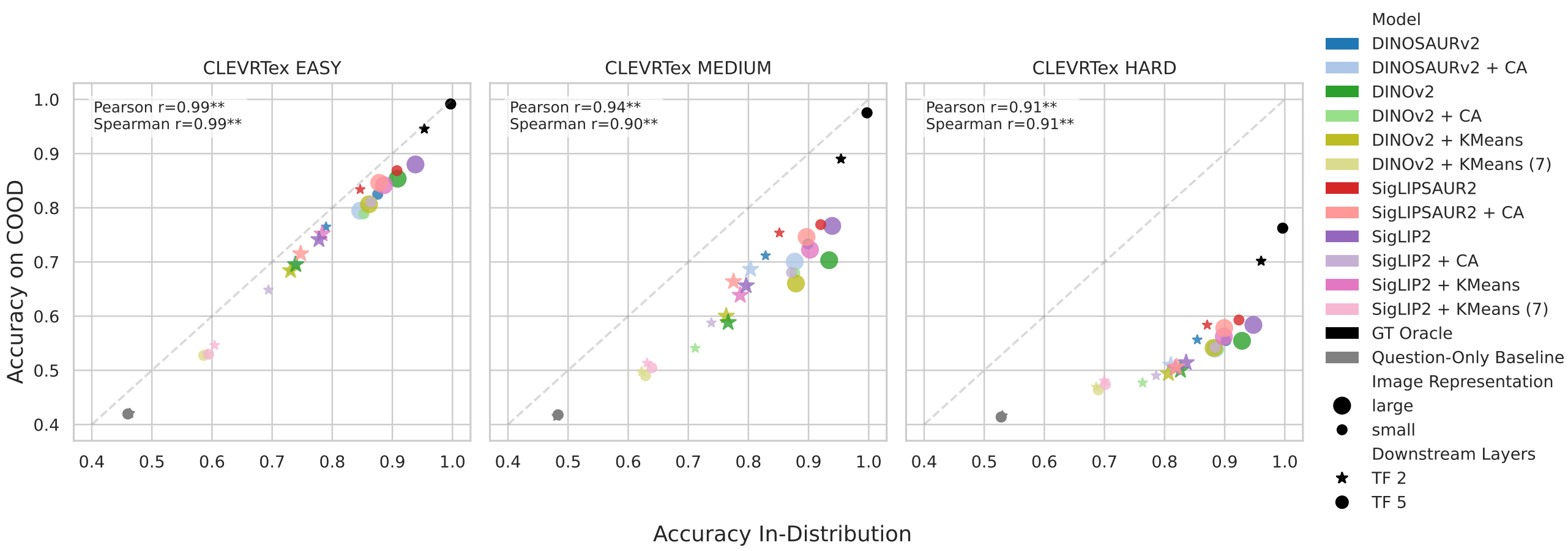

*Figure 11.* ID and COOD VQA accuracies are strongly correlated (Pearson and Spearman; $p < 0.01$). End-of-training results for CLEVRTex (easy, medium, hard) across all image representations and downstream models. The ground-truth oracle is in the top right (black) and the question-only baseline in the bottom left (gray).

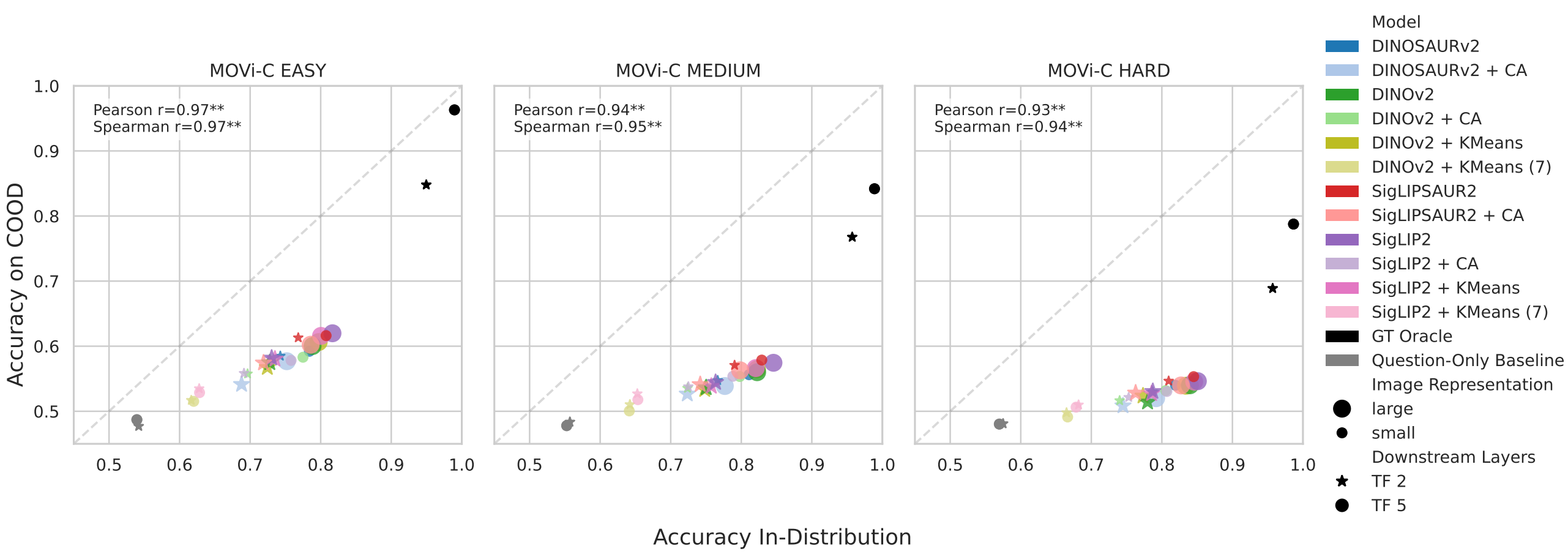

*Figure 12.* ID and COOD VQA accuracies are strongly correlated (Pearson and Spearman; $p < 0.01$). End-of-training results for MOVi-C (easy, medium, hard) across all image representations and downstream models. The ground-truth oracle is in the top right (black) and the question-only baseline in the bottom left (gray).

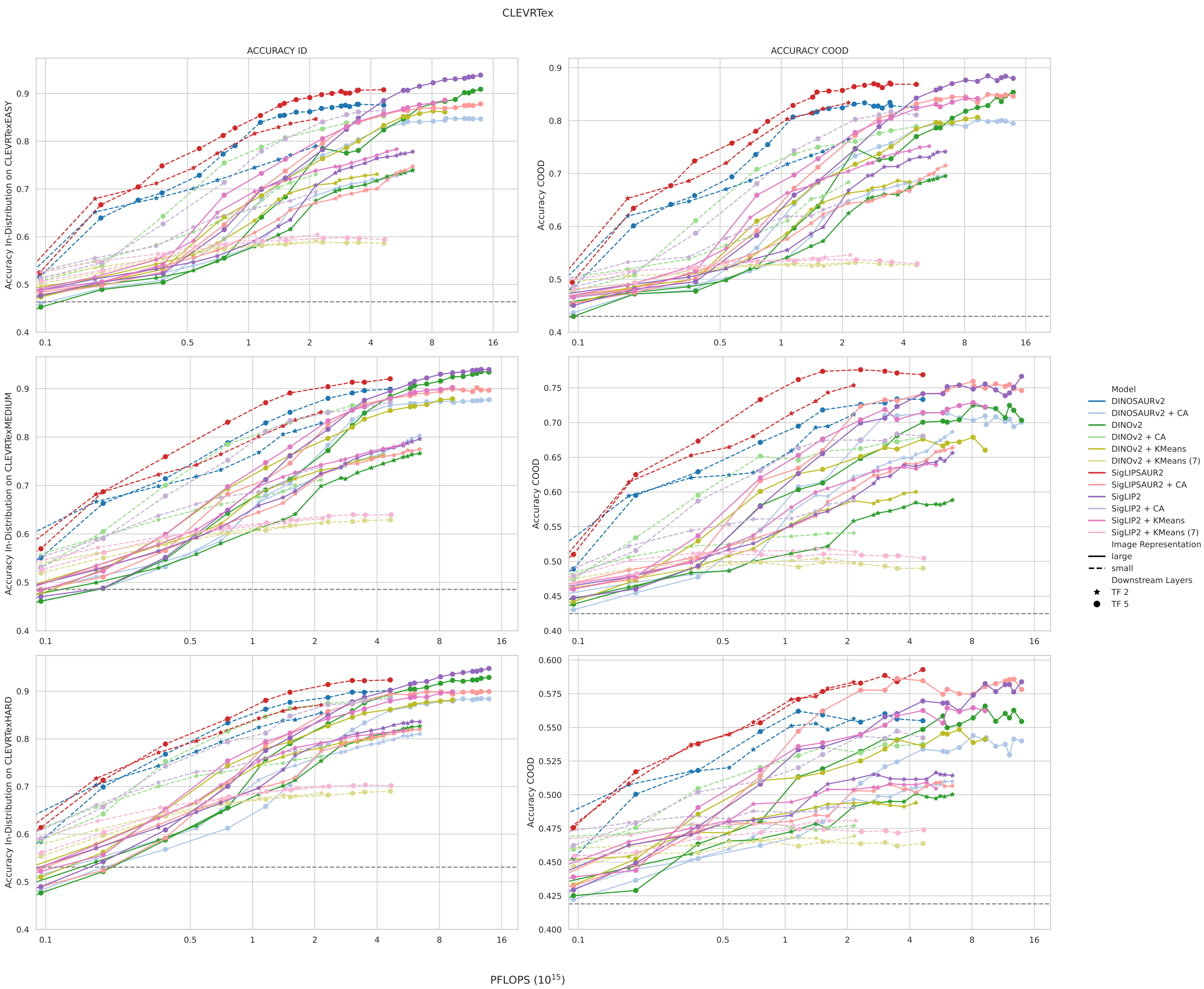

*Figure 13.* VQA in-distribution and compositional out-of-distribution accuracy for all CLEVRTex dataset variants with question-only baseline (lower: dashed gray).

*Table 6.* Attributes for the image and question generation for MOVi-C.

| **Shape** (36) | **Size** (3) |
|---|---|
| athletic shoe | small |
| boot | medium |
| sandal | large |
| flip flop | |
| ballet flat | |
| action figure | |
| stuffed animal | |
| board game | |
| puzzle toy | |
| construction toy | |
| toy vehicle | |
| musical toy | |
| doll | |
| game console | |
| ink cartridge | |
| computer mouse | |
| keyboard | |
| headphones | |
| router | |
| hard drive | |
| tablet | |
| coffee mug | |
| bowl | |
| plate | |
| hand towel | |
| screwdriver | |
| scissors | |
| tape roll | |
| plant pot | |
| lunch bag | |
| dish drying mat | |
| mixing bowl | |
| frying pan | |
| blender | |
| toaster | |
| coffee maker | |

*Table 7.* Hyperparameters of DINOSAURv2 and SigLIPSAUR2.

| **Hyperparameter** | | **DINOSAURv2** | **SigLIPSAUR2** |
|---|---|---|---|
| Training Steps | | 300k | 300k |
| Batch Size | | 128 | 128 |
| LR Warmup Steps | | 10k | 10k |
| Peak LR | | 0.0003 | 0.0002 |
| LR Schedule | | Cosine | Exp. Decay |
| Exp. Decay Half-Life | | - | 100k |
| Cosine T-Max | | 300k | - |
| Feature Extractor | | DINOv2_S | SigLIP2_B |
| Patch Size | | 14 | 16 |
| Feature Dim. | | 384 | 768 |
| Gradient Norm Clipping | | 0.1 | 0.1 |
| Image Size | | 224 | 224 |
| Cropping Strategy | | Full | Full |
| Image Tokens | | 256 | 196 |
| Decoder | Type | MLP | MLP |
| | Layers | 4 | 4 |
| | MLP Hidden Dim. | 2048 | 2048 |
| Slot Attention | Iterations | 3 | 3 |
| | Number of Slots | 7 | 7 |
| | Slot Dim. | 256 | 256 |
| | MLP Hidden Dim. | 1024 | 1024 |

*Table 8.* Number of steps for the two downstream models for all image representations.

| | | **Number of steps** | |
|---|---|---|---|
| **Model Name** | **Image Repr. Size** | **TF 2** | **TF 5** |
| SigLIP2 / SigLIPSAUR2 + CA | [196,768] | 641123 | 688733 |
| DINOv2 / DINOSAURv2 + CA | [256,384] | 600000 | 600000 |
| SigLIP2 + k-means | [128,768] | 718928 | 617749 |
| DINOv2 + k-means | [128,384] | 643602 | 652251 |
| SigLIPSAUR2 / DINOSAURv2 / SigLIP2 + CA / DINOv2 + CA / SigLIP2 + k-means (7) / DINOv2 + k-means (7) | [7,256] | 762329 | 809568 |

*Table 9.* VQA accuracy (%) of both downstream models (TF 2 & 5) on the compositional generalization test sets of MOVi-C with OOD background.

| | | **MOVi-C** | | |
|---|---|---|---|---|
| | Model | EASY | MEDIUM | HARD |
| **TF 2** | DINOv2 | 57.7 | 53.7 | 51.6 |
| | DINOSAURv2 | 58.3 | 55.0 | 53.2 |
| | SigLIP2 | 58.3 | 54.6 | 53.2 |
| | SigLIPSAUR2 | 61.4 | 57.1 | 54.9 |
| **TF 5** | DINOv2 | 60.1 | 56.3 | 54.3 |
| | DINOSAURv2 | 59.3 | 56.0 | 54.4 |
| | SigLIP2 | 61.6 | 57.8 | 55.1 |
| | SigLIPSAUR2 | 61.4 | 57.9 | 55.5 |

*Table 10.* Object-centric model ablations measured by VQA compositional out-of-distribution downstream model accuracy for DINOSAURv2.

| | Model | #slots | CLEVRTex EASY | CLEVRTex MEDIUM | CLEVRTex HARD |
|---|---|---|---|---|---|
| TF 2 | DINOSAURv2 | 5 | 68.2 | 63.6 | 51.2 |
| TF 2 | DINOSAURv2 | 7 (default) | 76.5 | 71.2 | 55.6 |
| TF 2 | DINOSAURv2 | 9 | 80.0 | 72.4 | 55.1 |
| TF 2 | DINOSAURv2 | 11 | 79.7 | 73.6 | 56.3 |
| TF 5 | DINOSAURv2 | 5 | 71.9 | 65.3 | 51.0 |
| TF 5 | DINOSAURv2 | 7 (default) | 82.5 | 73.3 | 55.5 |
| TF 5 | DINOSAURv2 | 9 | 83.6 | 74.1 | 56.0 |
| TF 5 | DINOSAURv2 | 11 | 85.7 | 74.3 | 56.8 |

*Table 11.* Downstream model ablations for VQA compositional out-of-distribution accuracy for DINOv2-based representations.

| Model | Downstream Model #layers | $d_{model}$ | $d_{hidden}$ | CLEVRTex EASY | CLEVRTex MEDIUM | CLEVRTex HARD |
|---|---|---|---|---|---|---|
| DINOv2 | TF 5 (default) | 128 | 128 | 85.4 | 70.3 | 55.4 |
| DINOSAURv2 | TF 5 (default) | 128 | 128 | 82.5 | 73.3 | 55.5 |
| DINOv2 | TF 10 | 128 | 128 | 84.9 | 70.0 | 55.2 |
| DINOSAURv2 | TF 10 | 128 | 128 | 81.3 | 70.7 | 53.5 |
| DINOv2 | TF 15 | 128 | 128 | 83.8 | 68.1 | 53.7 |
| DINOSAURv2 | TF 15 | 128 | 128 | 81.6 | 71.9 | 54.3 |
| DINOv2 | TF 5 | 128 | 256 | 81.0 | 71.4 | 52.2 |
| DINOSAURv2 | TF 5 | 128 | 256 | 81.1 | 72.6 | 53.8 |
| DINOv2 | TF 5 | 128 | 512 | 81.5 | 71.0 | 53.8 |
| DINOSAURv2 | TF 5 | 128 | 512 | 81.7 | 69.8 | 53.7 |
| DINOv2 | TF 5 | 256 | 1024 | 85.3 | 70.1 | 56.2 |
| DINOSAURv2 | TF 5 | 256 | 1024 | 81.1 | 68.2 | 54.5 |

*Table 12.* VQA Accuracy in-distribution for all image representations and the small downstream model (TF 2).

| | CLEVRTex | | | Super-CLEVR | | | MOVi-C | | |
|---|---|---|---|---|---|---|---|---|---|
| **TF 2** | E | M | H | E | M | H | E | M | H |
| Question-Only Baseline | 46.3 | 48.1 | 53.0 | 49.0 | 50.7 | 59.5 | 54.2 | 55.7 | 57.5 |
| DINOv2 | 73.9 | 76.6 | 82.6 | 65.9 | 68.2 | 81.2 | 72.8 | 75.0 | 78.0 |
| DINOv2 + CA | 73.1 | 71.2 | 76.3 | 65.5 | 66.7 | 79.4 | 69.6 | 72.4 | 74.0 |
| DINOv2 + KMeans (7) | 59.2 | 62.2 | 68.6 | 55.5 | 58.0 | 70.9 | 61.7 | 64.2 | 66.5 |
| DINOv2 + KMeans | 73.1 | 76.3 | 80.6 | 65.1 | 67.7 | 80.5 | 72.4 | 74.8 | 77.3 |
| DINOSAURv2 | 79.0 | 82.9 | 85.4 | 67.6 | 70.2 | 81.8 | 74.3 | 76.7 | 78.7 |
| DINOSAURv2 + CA | 73.8 | 80.4 | 81.0 | 63.0 | 65.6 | 78.4 | 68.8 | 72.3 | 74.5 |
| SigLIP2 | 77.8 | 79.6 | 83.6 | 68.1 | 69.3 | 82.5 | 73.1 | 76.4 | 78.7 |
| SigLIP2 + CA | 69.4 | 73.8 | 78.6 | 65.4 | 67.0 | 80.3 | 69.1 | 72.5 | 75.3 |
| SigLIP2 + KMeans (7) | 60.4 | 63.1 | 70.0 | 57.2 | 60.1 | 72.6 | 62.8 | 65.3 | 68.2 |
| SigLIP2 + KMeans | 78.3 | 78.6 | 81.8 | 66.9 | 70.1 | 81.8 | 73.5 | 75.8 | 78.5 |
| SigLIPSAUR2 | 84.7 | 85.2 | 87.1 | 71.0 | 73.4 | 84.1 | 76.8 | 79.0 | 81.0 |
| SigLIPSAUR2 + CA | 74.8 | 77.5 | 82.0 | 66.6 | 68.4 | 80.5 | 71.9 | 74.2 | 76.3 |
| GT Oracle | 95.3 | 95.4 | 96.1 | 91.3 | 91.6 | 93.5 | 95.0 | 95.7 | 95.7 |

*Table 13.* VQA Accuracy compositional out-of-distribution for all image representations and the small downstream model (TF 2).

| | CLEVRTex | | | Super-CLEVR | | | MOVi-C | | |
|---|---|---|---|---|---|---|---|---|---|
| **TF 2** | E | M | H | E | M | H | E | M | H |
| Question-Only Baseline | 42.1 | 41.5 | 41.6 | 45.0 | 45.2 | 44.1 | 47.7 | 48.4 | 48.1 |
| DINOv2 | 69.5 | 58.8 | 50.0 | 60.9 | 57.0 | 49.7 | 57.5 | 53.6 | 51.4 |
| DINOv2 + CA | 68.4 | 54.1 | 47.7 | 59.8 | 55.8 | 48.9 | 55.8 | 53.6 | 51.7 |
| DINOv2 + KMeans (7) | 53.0 | 49.8 | 46.9 | 50.8 | 49.9 | 47.5 | 51.7 | 51.0 | 49.8 |
| DINOv2 + KMeans | 68.4 | 60.0 | 49.4 | 59.5 | 56.2 | 49.6 | 56.7 | 53.4 | 52.4 |
| DINOSAURv2 | 76.5 | 71.2 | 55.6 | 60.6 | 58.6 | 50.9 | 58.5 | 54.7 | 53.0 |
| DINOSAURv2 + CA | 69.6 | 68.7 | 51.0 | 57.7 | 55.1 | 49.4 | 54.1 | 52.7 | 50.8 |
| SigLIP2 | 74.1 | 65.6 | 51.4 | 62.3 | 57.6 | 50.4 | 58.2 | 54.4 | 53.1 |
| SigLIP2 + CA | 64.8 | 58.8 | 49.0 | 60.4 | 56.5 | 49.5 | 55.8 | 53.8 | 52.1 |
| SigLIP2 + KMeans (7) | 54.6 | 51.4 | 48.1 | 52.0 | 51.6 | 48.5 | 53.5 | 52.7 | 51.0 |
| SigLIP2 + KMeans | 75.2 | 63.9 | 50.5 | 61.6 | 59.1 | 49.9 | 58.0 | 53.8 | 52.7 |
| SigLIPSAUR2 | 83.4 | 75.4 | 58.4 | 63.8 | 61.3 | 52.1 | 61.3 | 57.1 | 54.7 |
| SigLIPSAUR2 + CA | 71.5 | 66.4 | 50.7 | 60.5 | 56.4 | 49.9 | 57.4 | 54.1 | 52.8 |
| GT Oracle | 94.6 | 89.0 | 70.2 | 89.9 | 86.8 | 67.3 | 84.8 | 76.8 | 68.9 |

*Table 14.* VQA Accuracy in-distribution for all image representations and the big downstream model (TF 5).

| | CLEVRTex | | | Super-CLEVR | | | MOVi-C | | |
|---|---|---|---|---|---|---|---|---|---|
| **TF 5** | E | M | H | E | M | H | E | M | H |
| Question-Only Baseline | 46.0 | 48.3 | 52.8 | 48.9 | 50.9 | 59.9 | 53.9 | 55.3 | 57.0 |
| DINOv2 | 90.9 | 93.4 | 92.9 | 76.2 | 79.0 | 85.2 | 78.8 | 82.2 | 84.0 |
| DINOv2 + CA | 85.3 | 87.7 | 88.9 | 73.7 | 77.9 | 85.2 | 77.5 | 79.8 | 80.6 |
| DINOv2 + KMeans (7) | 58.6 | 62.9 | 69.0 | 55.0 | 58.2 | 70.8 | 62.0 | 64.1 | 66.6 |
| DINOv2 + KMeans | 86.1 | 87.9 | 88.2 | 71.7 | 77.1 | 85.5 | 79.7 | 82.2 | 83.4 |
| DINOSAURv2 | 87.6 | 89.9 | 90.2 | 73.5 | 77.9 | 87.3 | 78.4 | 81.1 | 81.9 |
| DINOSAURv2 + CA | 84.7 | 87.7 | 88.4 | 70.3 | 74.6 | 85.9 | 75.2 | 77.7 | 79.2 |
| SigLIP2 | 93.8 | 93.9 | 94.8 | 80.6 | 84.9 | 91.7 | 81.7 | 84.6 | 85.1 |
| SigLIP2 + CA | 86.4 | 87.2 | 88.4 | 72.8 | 76.4 | 85.1 | 75.8 | 78.8 | 80.7 |
| SigLIP2 + KMeans (7) | 59.4 | 63.9 | 70.2 | 57.3 | 60.6 | 73.5 | 62.8 | 65.3 | 67.8 |
| SigLIP2 + KMeans | 88.6 | 90.2 | 89.9 | 74.9 | 80.5 | 87.9 | 80.0 | 82.1 | 84.6 |
| SigLIPSAUR2 | 90.8 | 92.0 | 92.4 | 77.5 | 81.3 | 89.0 | 80.8 | 82.9 | 84.5 |
| SigLIPSAUR2 + CA | 87.8 | 89.7 | 89.9 | 76.0 | 80.8 | 88.9 | 78.6 | 79.8 | 82.7 |
| GT Oracle | 99.7 | 99.7 | 99.6 | 95.8 | 97.0 | 97.1 | 99.0 | 98.9 | 98.7 |

*Table 15.* VQA Accuracy compositional out-of-distribution for all image representations and the big downstream model (TF 5).

| | CLEVRTex | | | Super-CLEVR | | | MOVi-C | | |
|---|---|---|---|---|---|---|---|---|---|
| **TF 5** | E | M | H | E | M | H | E | M | H |
| Question-Only Baseline | 42.0 | 41.8 | 41.4 | 45.0 | 44.9 | 43.9 | 48.7 | 47.8 | 48.0 |
| DINOv2 | 85.4 | 70.3 | 55.4 | 68.1 | 63.0 | 51.7 | 60.0 | 56.0 | 54.0 |
| DINOv2 + CA | 78.9 | 67.9 | 53.8 | 65.2 | 61.1 | 50.8 | 58.3 | 55.3 | 53.2 |
| DINOv2 + KMeans (7) | 52.7 | 49.0 | 46.4 | 50.6 | 49.9 | 47.2 | 51.5 | 50.1 | 49.1 |
| DINOv2 + KMeans (128) | 80.6 | 66.0 | 54.1 | 64.4 | 61.8 | 51.3 | 60.6 | 56.5 | 53.9 |
| DINOSAURv2 | 82.5 | 73.3 | 55.5 | 64.6 | 60.8 | 52.1 | 59.2 | 55.6 | 54.0 |
| DINOSAURv2 + CA | 79.5 | 70.1 | 54.0 | 62.8 | 59.3 | 51.1 | 57.7 | 53.9 | 52.0 |
| SigLIP2 | 88.0 | 76.7 | 58.4 | 70.6 | 66.8 | 54.3 | 62.0 | 57.5 | 54.6 |
| SigLIP2 + CA | 81.1 | 68.1 | 54.2 | 64.7 | 60.9 | 51.5 | 57.8 | 55.4 | 53.0 |
| SigLIP2 + KMeans (7) | 53.0 | 50.5 | 47.4 | 52.4 | 50.9 | 48.1 | 52.9 | 51.8 | 50.6 |
| SigLIP2 + KMeans (128) | 84.2 | 72.3 | 56.2 | 66.3 | 63.7 | 52.9 | 61.6 | 56.6 | 54.5 |
| SigLIPSAUR2 | 86.9 | 76.9 | 59.3 | 66.8 | 62.4 | 53.5 | 61.6 | 57.9 | 55.3 |
| SigLIPSAUR2 + CA | 84.6 | 74.6 | 57.8 | 65.6 | 62.1 | 52.7 | 60.2 | 56.3 | 54.0 |
| GT Oracle | 99.2 | 97.5 | 76.2 | 93.3 | 91.6 | 67.0 | 96.3 | 84.2 | 78.8 |

*Table 16.* VQA accuracy (%) of both downstream models (TF 2 & 5) on the respective compositional generalization test sets for all models, trained on easy (E), medium (M), and hard (H) training sets. We compute deltas compared to the original pretrained vision encoder.

| | CLEVRTex | | | Super-CLEVR | | | MOVi-C | | |
|---|---|---|---|---|---|---|---|---|---|
| **TF 2** | E | M | H | E | M | H | E | M | H |
| DINOv2 | 69.5 | 58.8 | 50.0 | 60.9 | 57.0 | 49.7 | 57.5 | 53.6 | 51.4 |
| DINOv2 + CA | -1.2 | -4.8 | -2.3 | -1.2 | -1.2 | -0.8 | -1.7 | -0.1 | 0.2 |
| DINOv2 + KMeans (7) | -16.5 | -9.1 | -3.1 | -10.1 | -7.1 | -2.2 | -5.8 | -2.6 | -1.6 |
| DINOv2 + KMeans | -1.1 | 1.2 | -0.6 | -1.4 | -0.8 | -0.1 | -0.7 | -0.3 | 0.9 |
| DINOSAURv2 | 7.0 | 12.3 | 5.6 | -0.3 | 1.6 | 1.2 | 1.0 | 1.1 | 1.6 |
| DINOSAURv2 + CA | 0.1 | 9.8 | 1.0 | -3.3 | -1.9 | -0.4 | -3.3 | -1.0 | -0.6 |
| SigLIP2 | 74.1 | 65.6 | 51.4 | 62.3 | 57.6 | 50.4 | 58.2 | 54.4 | 53.1 |
| SigLIP2 + CA | -9.3 | -6.9 | -2.4 | -1.9 | -1.0 | -0.9 | -2.3 | -0.6 | -0.9 |
| SigLIP2 + KMeans (7) | -19.5 | -14.3 | -3.3 | -10.3 | -6.0 | -1.9 | -4.7 | -1.7 | -2.1 |
| SigLIP2 + KMeans | 1.0 | -1.8 | -1.0 | -0.6 | 1.5 | -0.5 | -0.2 | -0.6 | -0.3 |
| SigLIPSAUR2 | 9.2 | 9.7 | 6.9 | 1.5 | 3.7 | 1.8 | 3.1 | 2.6 | 1.6 |
| SigLIPSAUR2 + CA | -2.6 | 0.8 | -0.8 | -1.8 | -1.2 | -0.5 | -0.7 | -0.3 | -0.2 |
| **TF 5** | E | M | H | E | M | H | E | M | H |
| DINOv2 | 85.4 | 70.3 | 55.4 | 68.1 | 63.0 | 51.7 | 60.0 | 56.0 | 54.0 |
| DINOv2 + CA | -6.5 | -2.4 | -1.7 | -2.9 | -1.9 | -0.9 | -1.7 | -0.7 | -0.8 |
| DINOv2 + KMeans (7) | -32.6 | -21.3 | -9.1 | -17.5 | -13.1 | -4.5 | -8.5 | -6.0 | -4.9 |
| DINOv2 + KMeans | -4.7 | -4.3 | -1.3 | -3.7 | -1.2 | -0.4 | 0.6 | 0.5 | -0.1 |
| DINOSAURv2 | -2.9 | 3.0 | 0.1 | -3.5 | -2.2 | 0.4 | -0.8 | -0.4 | -0.0 |
| DINOSAURv2 + CA | -5.9 | -0.3 | -1.4 | -5.3 | -3.7 | -0.6 | -2.3 | -2.1 | -2.0 |
| SigLIP2 | 88.0 | 76.7 | 58.4 | 70.6 | 66.8 | 54.3 | 62.0 | 57.5 | 54.6 |
| SigLIP2 + CA | -6.9 | -8.6 | -4.2 | -5.9 | -5.9 | -2.8 | -4.2 | -2.1 | -1.6 |
| SigLIP2 + KMeans (7) | -35.0 | -26.2 | -11.0 | -18.2 | -15.8 | -6.2 | -9.1 | -5.7 | -4.0 |
| SigLIP2 + KMeans | -3.8 | -4.4 | -2.1 | -4.3 | -3.1 | -1.4 | -0.4 | -0.8 | -0.1 |
| SigLIPSAUR2 | -1.1 | 0.2 | 0.9 | -3.8 | -4.4 | -0.8 | -0.4 | 0.4 | 0.7 |
| SigLIPSAUR2 + CA | -3.4 | -2.0 | -0.6 | -5.0 | -4.7 | -1.6 | -1.8 | -1.2 | -0.7 |

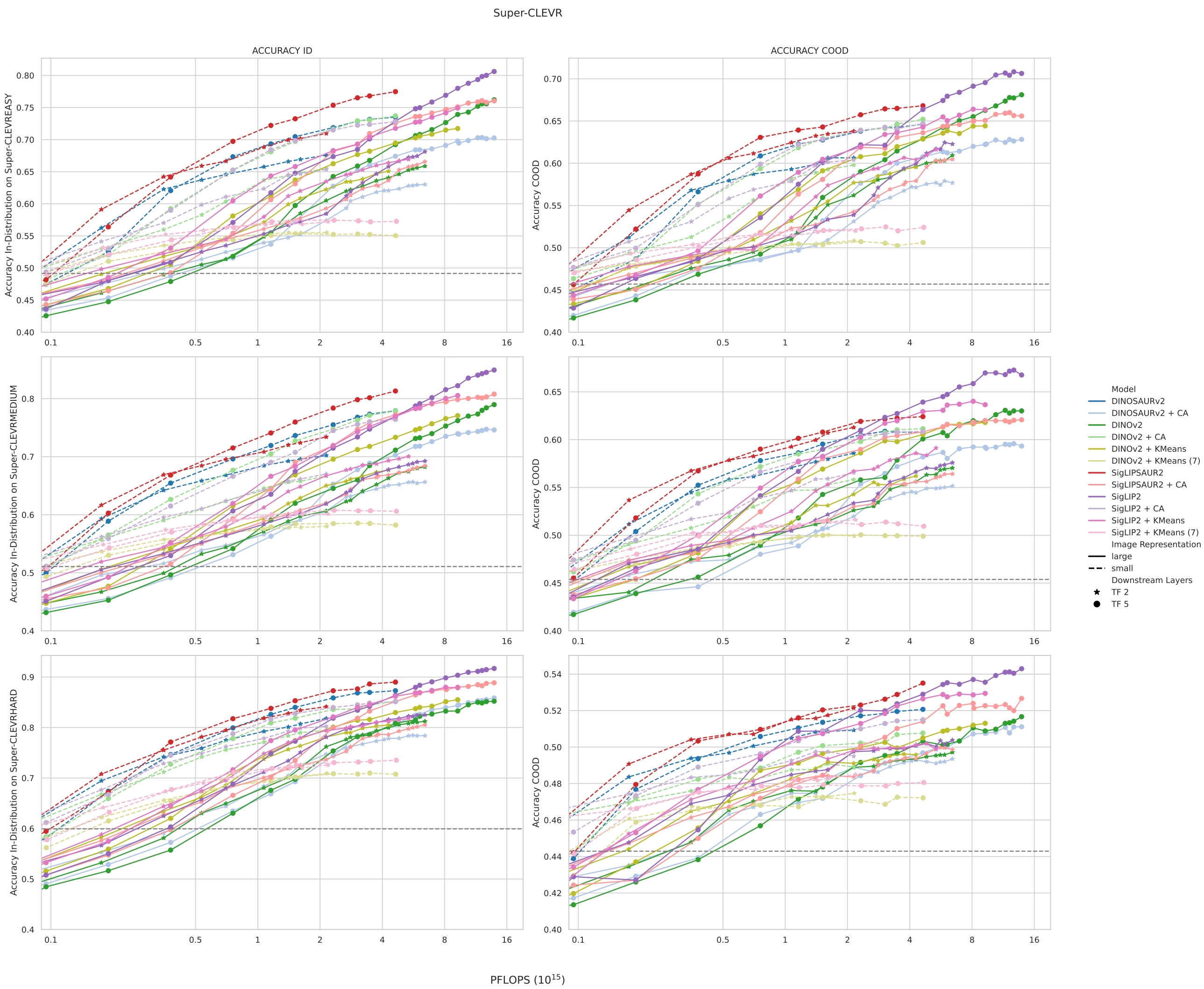

*Figure 14.* VQA in-distribution and compositional out-of-distribution accuracy for all Super-CLEVR dataset variants with question-only baseline (lower: dashed gray).

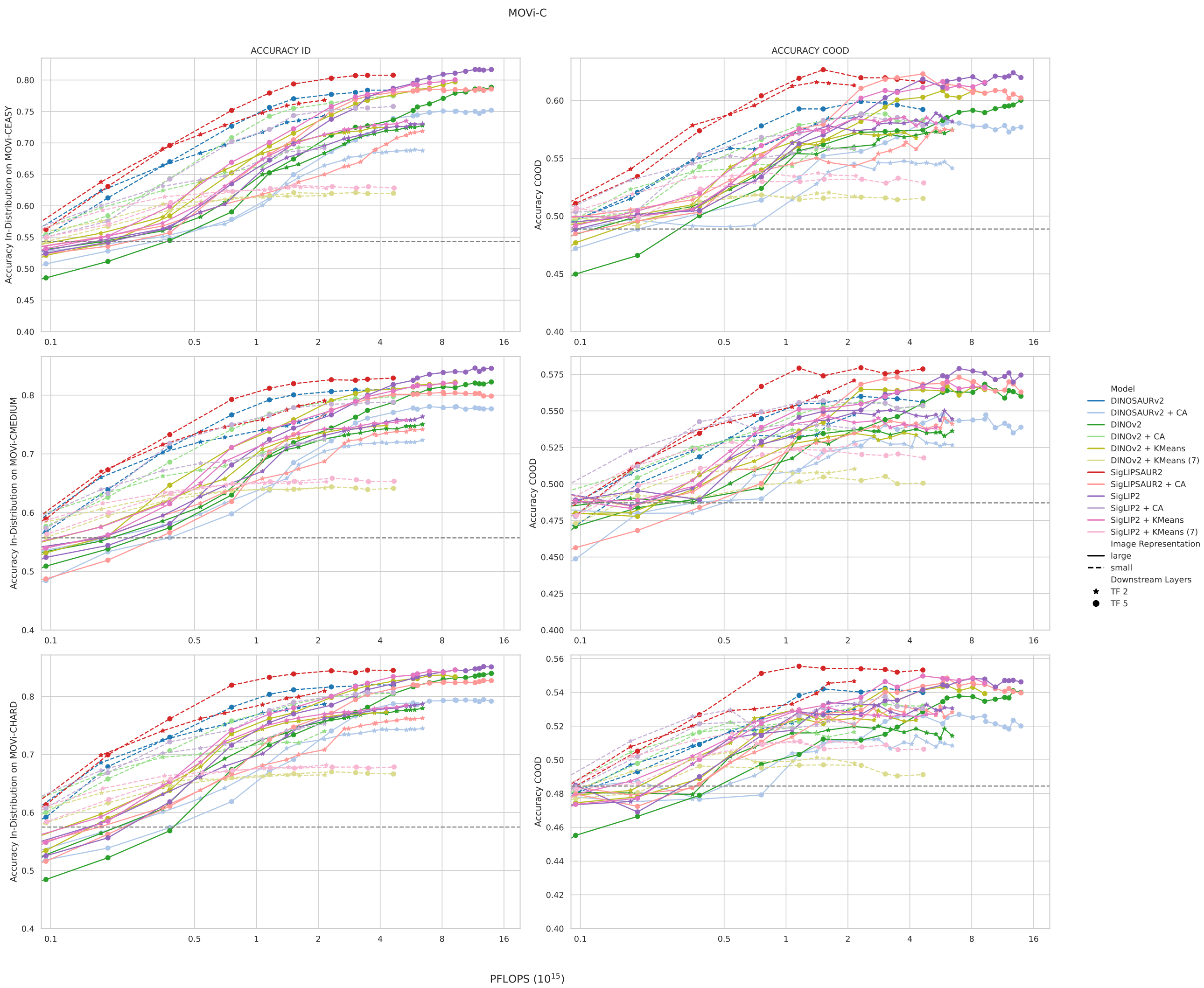

*Figure 15.* VQA in-distribution and compositional out-of-distribution accuracy for all MOVi-C dataset variants with question-only baseline (lower: dashed gray).

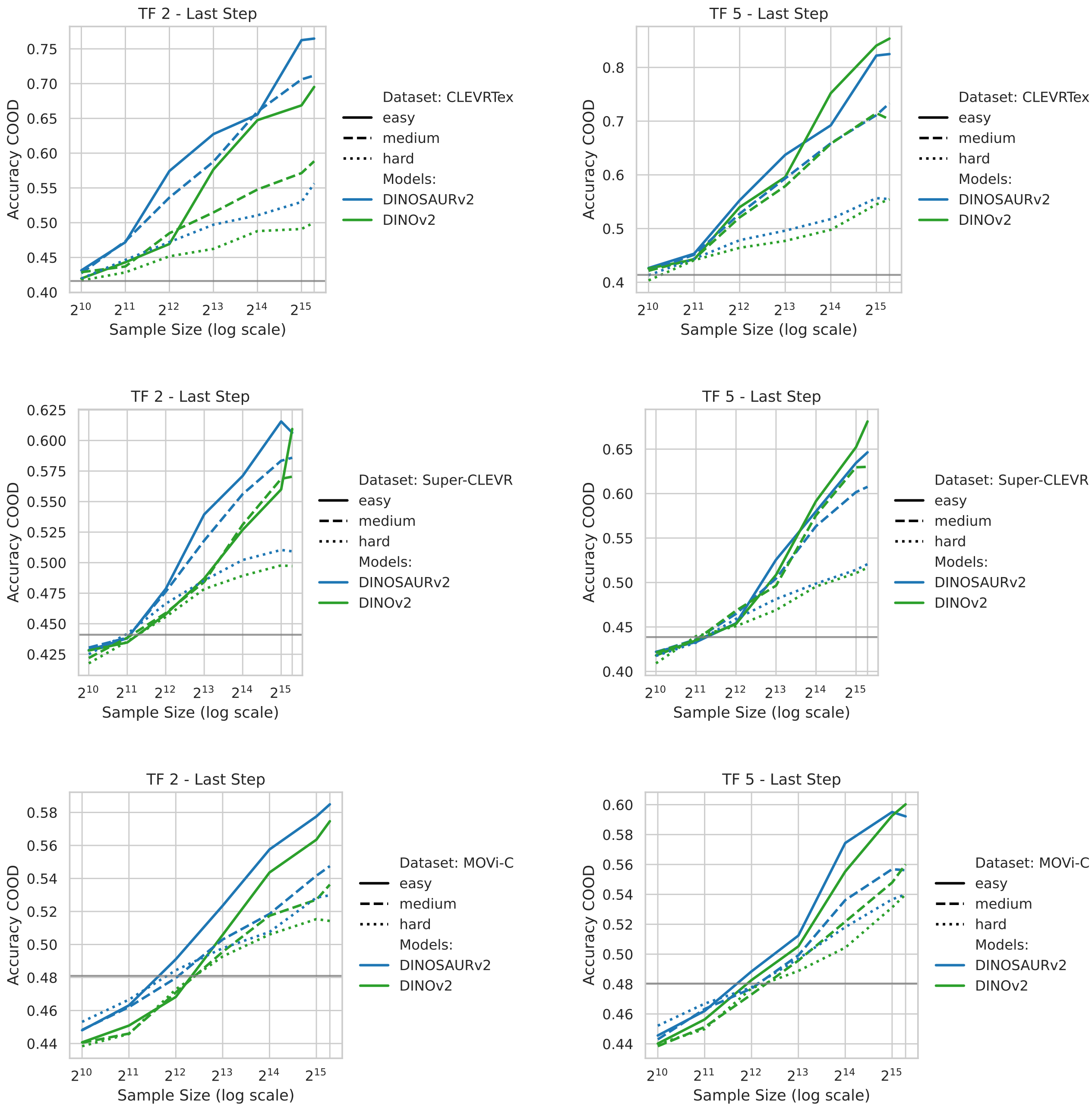

*Figure 16.* Compositional generalization of models trained on different subsets of the full data for CLEVRTex, Super-CLEVR and MOVi-C for DINOv2-based models.

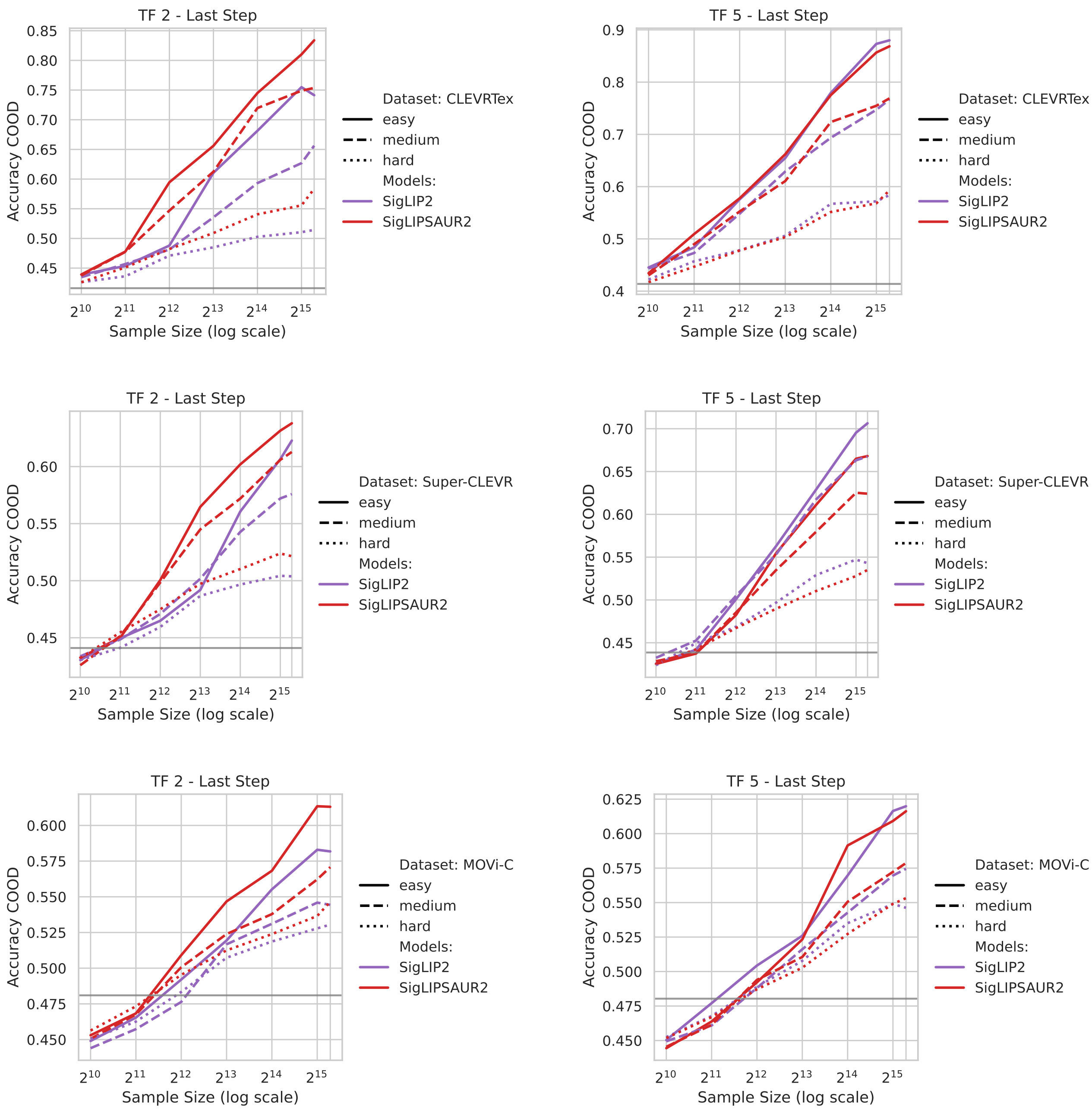

*Figure 17.* Compositional generalization of models trained on different subsets of the full data for CLEVRTex, Super-CLEVR and MOVi-C for SigLIP2-based models.