# OpenReview forum: "Are Object-Centric Representations Better at Compositional Generalization?"
_ICML.cc/2026/Conference — ICML 2026 regular_

### Official Review · Reviewer_7aqD · 2026-03-09

**Soundness:** 3
**Presentation:** 4
**Significance:** 3
**Originality:** 3
**Overall Recommendation:** 5
**Confidence:** 3

**Summary:**

The paper asks whether object-centric (i.e. slot-based) visual representation provide stronger compositional generalization that dense (patch-token) representations. To tackle this question, the authors introduce a VQA benchmarks, including visually rich, but controlled scenes, spanning CLEVRTex, Super-CLEVR, MOVi-C. In these benchmarks, compositional OOD (COOD) corresponds to holding out 20% of object-property combinations and varying the training diversity (i.e. easy/medium/hard) by restricting which combinations appear during training. The authors compare dense fondations encoders (DINOv2, SigLIP2) against object-centric counterparts (i.e. Slot Attention bottleneck training such as DINOSAURv2/SigLIPSAURv2). Evaluation is performed via a downstream transformer VQA model (with 2 different sizes). Additional controls are included for representation size, downstream compute, sample size, and alternative token reduction (k_means, cross-attention resizing). The main findings are: i) Object-centric representations win in harder COOD regimes (i.e. more constrained data and compute), ii) dense representation can catch-up (or even surpass) on easier regime with higher won stream compute and more diverse data.

**Compliance With Llm Reviewing Policy:**

Affirmed.

**Final Justification:**

I am willing to raise my score based on the additional uncertainty quantification provided in the response to Reviewer GefW. Although I still have some reservations about the interpretation of the object-centric advantage, I believe the rebuttal sufficiently strengthens the paper’s empirical support, and I now think it should be accepted at ICML.

**Key Questions For Authors:**

1. Is the OC benefit still present when controlling for token budget with a learned dense tokenizer? E.g., compare Slot Attention vs a learned tokenization/compression module trained with the same reconstruction loss but without slot competition/assignment.

2. How robust are results to the number of slots? You mention more slots can help COOD slightly; does the OC advantage disappear if you match OC token count to a dense reduced token count (e.g., 32/64 learned tokens)?

3. Where does OC help most? Can you break down COOD performance by question type (attribute query vs counting vs relations) to support the claim that object structure is specifically beneficial for composition?

4. How significant are the results ? Could you include error bars (i.e. accross seed) in the results.

**Limitations:**

The authors correctly addressed the limitations

**Strengths And Weaknesses:**

Strenghts:
1. Well motivated question and good controlled testbed: The benchmark target a specific notion of compositionally (novel combination of seen object properties) and systematically varies difficulty by reducing training diversity while keeping a fixed COOD test set. The underlying scientific question is impactful.

2. Controls are numerous: The paper explicitly accounts for confounds that often dominate OC-vs-dense comparison (i.e. token count, token dimensionality, downstream model capacity, and compute) by resizing representations with cross attention when needed.

3. Good baselines: Including question-only model and a ground-truth oracle (i.e. true object properties as image-token) helps to interpret the dataset difficulties and show that COOD is non-trivial even with perfect symbolic inputs as diversity drops.

4. Clear experiments: The 4 findings (diversity, representation type, compute, and sample size) are cleanly presented and supported by experiments.

Wekanesses:
1. The better « Object-centric » COOD performance might partly reflect the compression/regularization, but not object decomposition per se: DINOSAURv2/SigLIPSAUR2 use 7 slots (max objects / background), which seems to be a very « aggressive » bottleneck given the 196-256 patch token. This may act as an information bottleneck that regularizes the downstream VQA model (especially in TF2 settings), independent of the abjectness. While to paper do include other token reduction baselines (k-means, cross-attention reduction), theses might be strong enough baseline to fully separate the ’slot structure’ from ‘learned, task relevant compression »

2. The pertaining objective is a reconstruction object (i.e. reconstruct dense features), not a direct compositionality objective: The OC models are trained to reconstruct frozen dense features. This is a sensible “OC-from-foundation” pipeline, but it also means OC inherits the inductive biases and failure modes of the dense encoder and the learned decoder. It would strengthen the interpretation to include at least one alternative OC pretraining variant (e.g., reconstruction in pixel space, masked modeling, or contrastive OC objectives), or to show that the key trends persist across several OC training objectives.
3. Compute accounting is not fully convincing: The paper states it ignores cross-attention resizing compute in some comparisons (to focus on representation size). That choice is defensible for controlled ablations, but it makes “matched FLOPs” less literal and could change conclusions in regimes where resizing layers are a non-trivial fraction of cost.

4. The evaluation is VQA-only, but compositional generalization might be task-dependent: VQA emphasizes relational reasoning and attribute binding, where object tokens plausibly help. It would be informative to add at least one additional downstream task (e.g., COOD classification of queried attributes, retrieval, or structured prediction) to test how general the OC advantage is.
5. Results significancy: There is not error bars, making it difficult to assess the significancy of the results (especially in the sample efficiency part).

---

> ### Author Rebuttal · Authors · 2026-03-31
>
> We thank the reviewer for the detailed and thoughtful review and address each concern below.
>
> ### W1: OC advantage: compression vs. decomposition
>
> We agree this is an important concern and designed our experiments to address it directly. The key comparison is with size-matched 7-token baselines: DINOSAURv2 (SA bottleneck), DINOv2 + CA (cross-attention to 7 learned queries, trained jointly), and DINOv2 + k-means(7) (7 cluster centroids, no training). These baselines capture both learned and unsupervised compression at the same token count and roughly the same bottleneck level.
>
> The CA baseline is trained with the same downstream objective as the main models, yet DINOSAURv2 shows a substantial and consistent advantage at equal token count (Table 1). This suggests that slot-based representations capture structurally different, more object-compositional information than dense representations under a comparable bottleneck. While fully disentangling compression from decomposition is difficult, the consistent pattern across these baselines points to the SA slot-competition mechanism as an important factor.
>
> ### W2: Only one OC pretraining objective
>
> Reconstruction of dense self-supervised features from a frozen encoder is a popular and well-validated OC pretraining paradigm (Seitzer et al. 2023, Mamaghan et al. 2024), and using it uniformly across all OC models ensures the only variable between OC and dense is representational structure, not training signal.
>
> ### W3: Compute accounting for cross-attention resizing
>
> We agree that proper accounting is important here. DINOv2 + CA adds relatively little overhead (learnable 7 query tokens), while DINOSAURv2 + CA has substantial compute overhead but, in any case, is not worth using, as it degrades performance compared to DINOSAURv2 (Table 1). We discuss this at the end of Section 4.2.
>
> ### W4: VQA-only evaluation
>
> We agree that evaluating additional downstream tasks would further strengthen the findings; for extra results on property prediction, please see our response to reviewer dVeu “Why VQA …?”.
>
> We also note that VQA covers multiple subtasks, including attribute queries, counting, spatial relations, and existence, thus already providing several evaluation dimensions within a single framework. See also our response to reviewer dVeu “Are OC representations …?”.
>
> ### W5/Q4: No error bars
>
> We agree on the importance of seed variance and refer to the answer to Q1 for the reviewer GefW regarding additional seeds on CLEVRTex.
>
> We additionally note that the full experimental grid required over 70k GPU-hours, making multi-seed replication of every condition impractical. However, the consistency of the main trends across three base datasets (CLEVRTex, Super-CLEVR, MOVi-C) and two foundation model families (DINOv2 and SigLIP2) already provides a form of cross-replication.
>
>
>
> ### Q1: Controlling token budget with a dense tokenizer
>
> We performed exactly this experiment: we replaced the Slot Attention bottleneck in DINOSAURv2 with a cross-attention module trained with the same dense-feature reconstruction loss but without slot competition or assignment. This yielded substantially worse results, as mentioned in Section 3.2. We refer to Wu et al. (2023) for a discussion of why cross-attention may be suboptimal as a pretraining bottleneck in this setting.
>
>
> ### Q2: Robustness to the number of slots
>
> We ablate the number of slots in Table 8, evaluating 5, 7, 9, and 11 slots. We find that 9–11 slots slightly improve COOD over the default of 7, and in all cases, the main conclusions hold. The additional compute from a few extra slots is negligible compared to the full DINOv2 representation (256 tokens), so the OC advantage at matched compute is preserved or widened.
>
> For 32 or 64 slots, increasing OC slots would only raise DINOSAURv2’s representation capacity, consistent with the 9–11 slot trend, whereas reducing DINOv2’s token count via cross-attention resizing consistently degrades performance relative to full 256-token representation (Table 1, DINOv2 + CA). Such a comparison would therefore be even more favorable to OC representations.
>
>
>
> ### Q3: COOD performance breakdown by question type
>
> The table below reports, for each question type, the fraction of question types with significance in which OC outperforms dense, and the mean COOD accuracy delta, aggregated across datasets and difficulty levels (* = significance by a two-sided binomial test). Overall, across all downstream models and datasets for common question types, OC representations perform better on (Query/Compare) Size and Count/Exist question types.
>
> | Question Type | OC better (%) | Mean Delta |
> | --- | --- | --- |
> | Query Size | 94% | +4.6* |
> | Compare Size | 78% | +4.7* |
> | Exist | 89% | +3.2* |
> | Count | 78% | +3.0* |
> | Count Equal | 78% | +1.9* |
> | Count Greater | 72% | +1.9* |
> | Compare Shape | 72% | +3.4* |
> | Count Less | 53% | +0.6 |
> | Query Shape | 36% | -1.5 |

---

> > ### Author Rebuttal · Reviewer_7aqD · 2026-04-01
> >
> > The rebuttal addresses my main concern reasonably well. In particular, the added clarification that the object-centric models outperform size-matched learned and unsupervised token-reduction baselines makes the interpretation stronger that the gain is not merely due to aggressive compression, but is related to slot-based structure. The responses on slot-count robustness and question-type breakdown are also helpful. My remaining reservations are mainly about scope—only one OC pretraining objective, VQA-only evaluation, and limited uncertainty quantification across seeds—rather than about the core empirical trend itself. Overall, the rebuttal increases my confidence in the paper, but does not substantially change my overall assessment beyond reinforcing a weak-accept recommendation.

---

> > > ### Author Response · Authors · 2026-04-02
> > >
> > > We appreciate the thoughtful follow-up. We would like to briefly highlight two points from our rebuttal that directly address the remaining reservations on scope.
> > >
> > > **On the VQA-only evaluation:** in our response to Reviewer dVeu we included property prediction results (Material, Shape, Size classification and position MSE) across three seeds on CLEVRTex easy/medium/hard. The OC advantage is consistent and more pronounced on harder settings, confirming that the findings extend beyond VQA to a different downstream task.
> > >
> > > **On uncertainty quantification:** in our response to Reviewer GefW we reported mean±std across three seeds for DINOv2 vs. DINOSAURv2 on all CLEVRTex splits. The gaps, especially on the medium setting, are stable across seeds, ruling out noise as the main factor.
> > >
> > > We understand and respect the concern about the single OC pretraining objective and will discuss it explicitly as a limitation and direction for future work. We are glad the rebuttal increased confidence in the core empirical trend.

---

### Official Review · Reviewer_Dzxw · 2026-03-13

**Soundness:** 3
**Presentation:** 3
**Significance:** 3
**Originality:** 2
**Overall Recommendation:** 3
**Confidence:** 3

**Summary:**

This paper studies whether object-centric (OC) representations provide a fundamental advantage for compositional generalization compared to standard dense vision representations. It introduces a VQA benchmark across three synthetic worlds (CLEVRTex, Super-CLEVR, and MOVi-C) to measure how models generalize to unseen combinations of familiar object properties, which includes a million-scale dataset and a training recipe. The study compares foundation models (DINOv2, SigLIP2) against their OC counterparts (DINOSAURv2, SigLIPSAUR2) while rigorously controlling for training data diversity, sample size, representation size, downstream model capacity, and compute. The evaluation reveals that OC models are better at harder compositional generalization cases and are more sample efficient under limited compute or extreme data scarcity, while original dense representations require more data and compute to achieve the same level under the same setting.

**Compliance With Llm Reviewing Policy:**

Affirmed.

**Final Justification:**

**Soundness.** The paper presents a careful and well-controlled empirical study, with thorough experiments supporting its claims. The rebuttal clarifies scaling behavior and improves interpretation.

**Originality.** The contribution is mainly empirical. The benchmark extends prior work in scale and richness, but the core design is not novel. The VQA design was also used as a protocol in evaluating visual representations in prior work.

**Significance.** The paper provides useful insights into when object-centric representations help (e.g., limited data/compute) and when dense models surpass them. However, the lack of new methods limits impact.

**Clarity.** The paper is clear and well-structured, with corrected claims after the rebuttal.

Regarding my concerns:

W1: Benchmark design is straightforward, and the VQA protocol is not a novel contribution.

W2: No new method is proposed.

W3: Scaling claims are now clearer and better aligned with results.

Overall, it is a solid empirical study with useful insights, but with limited novelty and significance.

Final recommendation: Weak reject.

**Key Questions For Authors:**

1. How were the hyperparameters chosen for the different representations? Do they share the same set of hyperparameters?

**Limitations:**

The paper did not discuss the limitation of OC representations or the scope of their benchmark. This can be improved by adding such discussion.

**Strengths And Weaknesses:**

Strengths:
1. The paper offers assets that could be useful in compositional generalization research, consisting of a dataset and a benchmark.
2. The comparison controls multiple factors in downstream training and studies their effects, which provides insights into the property of the dense representations and the OC representations in Section 4.

Weaknesses:
1. The benchmark design has limited novelty, and the paper does not show its advantage over existing synthetic datasets or benchmarks for compositional generalization.
2. The paper does not propose a novel method that further improves representations for compositional generalization. The evaluated OC models are proposed by previous work.
3. Although the paper claims OC representations generalize better compositionally under limited data, diversity, or compute, it does not have a better scaling property. From Table 1, their advantage becomes marginal or negative when paired with a larger downstream model (TF 5) on easier tasks, suggesting that the inductive bias of OC models is primarily beneficial under limited compute or extreme data scarcity. Based on Figure 3, DINOv2 seems to have a better scaling property, especially on easy and medium sets as DINOSAURv2 saturates earlier. Based on TF 5 of Figure 4, the DINOv2 and SigLIP2 surpass their OC counterparts at the right end.

---

> ### Author Rebuttal · Authors · 2026-03-31
>
> We thank the reviewer for the detailed feedback. We appreciate the recognition that the paper offers useful assets for compositional generalization research and that the controlled multi-factor comparison provides insights into the properties of dense and OC representations. We address the concerns below.
>
> ### W1: Limited novelty of the benchmark
> We respectfully disagree. The key contribution is a *controlled compositional train/test split design for multi-object, multi-attribute scenes* evaluated via a non-generative VQA protocol that can assess any vision representation. The closest prior works are strictly more limited:
>
> - **Kim et al. (2024)** frame the task as a *generative* problem (input-output image sequences), which precludes evaluation of general vision representations. Their scenes contain at most 2 objects (vs. our 3–6) and at most 192 attribute combinations (vs. our 2,688 in Super-CLEVR). Their most complex dataset, CLEVRTex, is visually our *simplest* world.
> - **Montero et al. (2024)** use *single-object* images, removing multi-object compositional reasoning entirely.
>
> Additionally, SigLIPSAUR2, an OC variant of SigLIP2, is itself a new model contribution.
>
> ### W2: No novel OC architecture
> Indeed, we do not propose a new OC architecture, and consider this a methodological strength rather than a weakness. The goal of the study is explicitly stated as *"a fair and systematic comparison of dense and object-centric representations"* (Contribution 2). Using established, competitive OC methods (DINOSAURv2, SigLIPSAUR2) under matched backbone, capacity, and compute conditions means that any observed difference in COOD performance can be attributed solely to OC representational bias, not to other factors.
>
> We believe that rigorously establishing whether and when existing OC methods provide a compositional generalization advantage is a necessary step before drawing conclusions about which architectural directions to pursue.
>
> ### W3: OC advantage vs Scaling
> We completely agree with the observations, but do not see a contradiction. The reviewer correctly observes that dense representations have better scaling properties, but this is not in contradiction with our main claim; it *is* our main claim, framed in terms of the upper limit rather than a constrained setting. Specifically, our abstract states: *"object-centric representations offer stronger compositional generalization when any one of dataset size, training data diversity, or downstream compute is constrained."* The flip side is that dense representations can match or surpass OC when none of these is constrained or enough.
>
> Concretely, mapping the reviewer's observations to our findings:
> - *"OC advantage becomes marginal or negative when paired with TF 5 on easier tasks"* This is exactly Finding 2: OC outperforms dense for harder settings and smaller downstream models; dense matches or surpasses them on easier tasks.
> - *"DINOv2 seems to have a better scaling property on easy and medium sets as DINOSAURv2 saturates earlier"* This is Finding 3: dense encoders surpass with sufficient diversity and compute.
> - *"DINOv2 and SigLIP2 surpass their OC counterparts at the right end of Figure 4"* This is Finding 4. The right end of Figure 4 is the easy setting with full data and TF 5, precisely the "ample resources" regime where we claim dense is better.
>
> The reviewer's reading is therefore entirely consistent with our findings.
>
> ### Q: Hyperparameter choices for different representations
> We follow common practice for each model family. For the OC models (DINOSAURv2, SigLIPSAUR2), we follow the hyperparameters from (Seitzer et al. 2023, Mamaghan et al. 2024, Didolkar et al. 2025): 7 slots (maximum number of objects + 1 background, a standard convention), SA iterations, and MLP decoder. We performed small ablations by training upstream and downstream combination, and used the best VQA validation performance as the upstream setting. For the downstream VQA model, we follow Mamaghan et al. (2024): learning rate 1e-4, batch size 128, 600k steps, T5-base for question encoding. *All representations share the same downstream hyperparameters*, so differences in COOD performance cannot be explained by differences in tuning effort. Ablations on downstream model capacity (varying the number of transformer layers and hidden dimension) are provided in Table 9, showing that conclusions are robust across a wide range of downstream model sizes.
>
> ### Limitations
> We agree that a dedicated limitations section would strengthen the paper. We will discuss the (1) reliance on synthetic data, which provides experimental control but limits immediate real-world applicability; (2) scope restricted to object-property composition (one notion of compositionality); (3) evaluation on three specific visual worlds, though their diversity mitigates this concern; and (4) SA as the primary OC mechanism, with cross-attention and k-means as non-SA alternatives already included.

---

> > ### Author Rebuttal · Reviewer_Dzxw · 2026-04-04
> >
> > Thank you for the detailed response which clarifies several points.
> >
> > - Regarding W1, I agree that extending compositional splits to more complex multi-object scenes and enabling evaluation via a non-generative VQA protocol are useful contributions. The multi-dataset setup and support for evaluating arbitrary vision representations are also valuable. However, I remain unconvinced that the split design is a novel contribution.
> >   -  The core idea—holding out attribute combinations and controlling training diversity—is well established in prior work (e.g., CLEVR-CoGenT), which already evaluates compositional generalization in multi-object VQA settings.
> >   -  In this work, (1) the split mechanism is unchanged, (2) the difficulty control via reducing combinations is straightforward, and (3) the differences lie primarily in scale and visual richness. These are meaningful but incremental extensions rather than conceptual novelty.
> >   -  Given that no novel method or clear insight/guideline to a method outperforming existing ones is proposed (W2), the overall contribution is primarily empirical. While the study is thorough and well-executed, I still view the main contributions as incremental.
> >
> > - Regarding W3, I now better understand the conclusions, but find the original wording somewhat imprecise and potentially overstated. In particular, the phrase "OC achieves higher COOD across compute budgets" in Finding 3 is misleading, as dense representations catch up to or surpass OC models substantially at higher compute especially in easier settings, according to the trends in Figure 3. Therefore, I suggest revising the claim to more precisely reflect the results.

---

> > > ### Author Response · Authors · 2026-04-04
> > >
> > > We thank the reviewer for their effort and careful consideration of our rebuttal. We appreciate the acknowledgement that the study is **"thorough and well-executed"**.
> > >
> > > ### W1: Novelty of the split design
> > > We agree, and want to clarify that we do not claim the split design is itself novel. The idea of holding out attribute combinations to test compositional generalization is indeed established in prior work, including CLEVR-CoGenT (Johnson et al. 2017). As the reviewer correctly summarizes, the main differences of the benchmark lie in (1) scale and visual richness, and (2) the VQA evaluation protocol, which enables evaluation of arbitrary vision representations. We will make this framing more explicit in the revision.
> > >
> > > ### W2: Contribution through systematic evaluation
> > > Rigorous empirical evaluation, without proposing novel architectures, has produced lasting contributions in machine learning: scaling law analyses and systematic empirical evaluations are widely recognized precisely because they established reliable trends about model behavior. This work provides analogous grounding for OC representations, characterizing when the OC inductive bias helps for compositional generalization, which we believe is a necessary foundation before architectural improvements can be properly motivated or evaluated.
> > >
> > > ### W3: Wording of Finding 3
> > > We agree that the current phrasing "OC achieves higher COOD across compute budgets" can be misleading. The tightened version of Finding 3 is: *"At constrained downstream compute, object-centric representations achieve higher COOD; dense representations can match or surpass them on easier settings with substantially more compute."* We will revise the finding box accordingly, and take the reviewer's concern about precise wording carefully into account throughout the revision.
> > >
> > > We kindly ask the reviewer to consider these clarifications when making their final decision.

---

### Official Review · Reviewer_dVeu · 2026-03-15

**Soundness:** 4
**Presentation:** 4
**Significance:** 3
**Originality:** 3
**Overall Recommendation:** 5
**Confidence:** 4

**Summary:**

This paper investigates how vision–language models (with and without explicit object-centric representations) generalize to unseen compositions of object properties. Compared to prior works, this paper aims to be more systematic. In order to do so, they create their own controlled dataset of VQA questions with synthetically generated images (using prior frameworks that can generate multiple object scenes). 20% of object combinations are held-out and models are trained on a proportion of the remaining images. Generalization difficulty is considered a function of training diversity (i.e. generalization is harder if the model is trained on a smaller subset of images). Visual representations from pre-trained encoders (DINOv2, SigLIP2) and object-centric variants (auto-encoders with a slot attention bottleneck) are used and compared. The downstream VQA model is a Transformer.

They find that generalization improves (overall) with more training diversity. And that object-centric representations offer better generalization with less diverse training data, equal training compute, and with more sample efficiency.

**Compliance With Llm Reviewing Policy:**

Affirmed.

**Final Justification:**

I am a proponent of this paper: it is very well written, thorough, focuses on an important problem, and presents interesting findings. The authors responded to my review and resolved any questions. I would be happy to maintain my score of Accept.

**Key Questions For Authors:**

1. I have a simple suggestion: compositional generalization is very broad (it is covered by many papers but poorly defined) so perhaps the authors can also engage with Hupkes 2019 ("Compositionality decomposed...") which offers a framework for aspects of compositional generalization and McCurdy 2024 ("Toward compositional behavior...") which offers a definition for compositional behavior.

2. There is an independent dimension in the presented data (beyond object–property combinations): we can also consider the syntactic structure of the VQA problems. For example, the CLEVRTex question in Fig. 1 has a problem structure like "shape(front(right({object}), filter(behind({object}), {size}))))". Do the authors think there is sufficient coverage in the training sets to ensure models will have seen such problem structures at test time? Since we only want to measure object–property generalization? Why not just test for object detection or classification, rather than VQA?

3. This paper presents findings in favor of object-centric biases for models learning object-centric tasks. Do the authors think object-centric representations are less performant on other tasks? Also, previous works (e.g. CLIP, DINO) have shown that visual representations can be improved (e.g. with scale, better self-supervision objectives) to outperform representations from models with task-specific inductive biases. How do the authors see their own work in that positioning?

**Limitations:**

Yes.

**Strengths And Weaknesses:**

This paper is both clear and systematic and should clearly be accepted.

**Soundness:** The comparisons in this paper are very thorough. A number of controlled experiments systematically vary data diversity, representation type, model size, sample size, training compute, etc. Trends are observed across three data sources (each has 50k images, 40 VQA instances per image). The task and metrics are simple and appear sound.

**Presentation:** This paper is extremely well written. The task setup, experimental design, and findings are very clearly expressed. It was very easy to read and understand. They engage with prior literature and explain the motivations (systematic experiments) for the new object-centric compositional generalization benchmark (over a few prior works).

**Significance:** There is great interest in the compositional capabilities of neural networks, including for pre-trained encoders and object-centric inductive biases for object–property generalization. This paper argues that prior works aren't sufficiently systematic and proposes an experimental framework to advance our understanding. I agree and believe this paper presents several clear results that help inform this field.

**Originality:** As expressed above, this work provides new findings and deepens our understanding of the generalization capabilities between models with/without object-centric biases. It introduces both new data and new experimental design to do so.

---

> ### Author Rebuttal · Authors · 2026-03-31
>
> We thank the reviewer for the very positive and thorough assessment. We are especially glad that the reviewer finds the paper "extremely well written" and the comparison "very thorough." We address each point below.
>
>
>
> ### Suggestion: Engage with Hupkes et al. (2019) and McCurdy et al. (2024)
>
> We thank the reviewer for the suggestion. In the taxonomy of Hupkes et al. (2019), our benchmark targets *systematicity*, generalization to novel recombinations of familiar object properties. McCurdy et al. (2024)'s broader behavioral framework encompasses this as a special case. We will clarify in the paper that our evidence is behavioral rather than mechanistic.
>
>
> ### Sufficient coverage of question structures in training?
>
> We agree that syntactic question structure is an independent dimension from object-property composition, and the reviewer is right to ask whether all structures seen at test time are also present during training.
>
> The answer is yes, by design. Our VQA questions are generated by the templated pipeline of Johnson et al. (2017), adapted for CLEVRTex and MOVi-C. Super-CLEVR uses the existing implementation of Li et al. (2023). Templates are applied uniformly to all images. The compositional holdout concerns exclusively *which specific objects (property combinations) appear*, not question templates or syntactic structures.
>
> We verified this directly: there are roughly 50 (MOVi-C), 70 (CLEVRTex), and 100 (Super-CLEVR) templates for each dataset. This results in at least over 17k (for Super-CLEVR) instantiations of each template in the training set of 40k training images.
>
>
>
> ### Why VQA rather than classification or object detection?
>
> VQA is a harder task that simultaneously tests the model's ability to handle multiple objects and their spatial and property relations within a single unified evaluation. Classification of a single attribute (e.g., "what color is this object?") tests only attribute binding in isolation. Object detection does not directly probe whether a model can reason about novel property combinations.
>
> For property prediction, we train a one-hidden-layer MLP on the representations of DINOv2 and DINOSAURv2, following the settings in Seitzer et al. (2023), for 3 seeds per CLEVRTex variant and report the mean±std below. The benefit of the object-centric DINOSAURv2 representation becomes clear on the harder generalizations (medium, hard), where the difference is especially large for Material and Shape classification. In summary, our main conclusions also hold for attribute classification.
>
> #### Property Prediction - COOD Test Set (mean±std, 3 seeds)
>
> | Model | CLEVRTex | Material (Acc) | Shape (Acc) | Size (Acc) | X (MSE) | Y (MSE) |
> | --- | --- | --- | --- | --- | --- | --- |
> | DINOSAURv2 | Easy | 0.888±0.000 | 0.780±0.002 | 0.887±0.006 | 0.004±0.000 | 0.001±0.000 |
> | DINOv2 | Easy | 0.883±0.019 | 0.814±0.068 | 0.913±0.030 | 0.019±0.005 | 0.004±0.001 |
> |  |  |  |  |  |  |  |
> | DINOSAURv2 | Medium | 0.773±0.002 | 0.755±0.002 | 0.842±0.006 | 0.005±0.000 | 0.001±0.000 |
> | DINOv2 | Medium | 0.634±0.039 | 0.643±0.019 | 0.832±0.022 | 0.044±0.002 | 0.007±0.000 |
> |  |  |  |  |  |  |  |
> | DINOSAURv2 | Hard | 0.607±0.006 | 0.618±0.016 | 0.675±0.014 | 0.009±0.000 | 0.002±0.000 |
> | DINOv2 | Hard | 0.473±0.067 | 0.493±0.013 | 0.707±0.034 | 0.061±0.004 | 0.010±0.000 |
>
>
>
>
> ### Are OC representations less performant on non-OC tasks?
>
> VQA is not exclusively an object-centric task: question types range from counting to relational reasoning (e.g., attribute comparisons across objects), and OC models perform competitively across all of them. More broadly, Yoon et al. (2023), Haramati et al. (2024), and Mamaghan et al. (2024) find OC models competitive or better on relational RL tasks and GQA.
>
> ### How does this work relate to scaling?
>
> At the upper end of our training diversity and downstream compute axis, the "easy" setting with TF 5 and full dataset, dense representations surpass their OC counterparts (Table 1, Fig. 3, Fig. 4). Our contribution is to systematically characterize *what the boundary looks like*: OC representations are preferable whenever any one of data diversity, sample size, or downstream compute is constrained, while dense representations become better when all three are ample.
>
> This positions our findings as complementary to scaling-focused work (Kempf et al. 2025, Uselis et al. 2025, Wiedemer et al. 2025): scaling dense representations can eventually overcome the OC inductive bias, but object-centric representations offer a more compute- and data-efficient path to strong compositional generalization in the constrained regime that practitioners often face.

---

> > ### Author Rebuttal · Reviewer_dVeu · 2026-04-03
> >
> > Thanks to the authors for their response! I am satisfied with their response and have checked the other reviews. I would like to maintain my original score, which I think is already good.

---

### Official Review · Reviewer_GefW · 2026-03-23

**Soundness:** 3
**Presentation:** 3
**Significance:** 3
**Originality:** 3
**Overall Recommendation:** 4
**Confidence:** 4

**Summary:**

This paper systematically evaluates the models based on object-centric representations in compositionality generalization tasks using fully controlled datasets. The results show that object-centric models perform better in more challenging compositional generalization settings, as well as under constraints such as limited sample size, reduced data diversity, or restricted downstream compute. In addition, the authors introduce a new VQA benchmark across three visually rich and controlled settings to assess how well models can reason about novel combinations of learned concepts. Overall, the work is solid, and the simulation experiments are well designed.

**Compliance With Llm Reviewing Policy:**

Affirmed.

**Final Justification:**

This paper presents solid work with systematic evaluations on synthetic datasets. However, the theoretical contribution is still somewhat limited without validation on realistic images, particularly in cases where forming object-centric representations becomes challenging. As such, I will maintain my original score.

**Key Questions For Authors:**

The DINSURv2 appears to achieve the largest performance gains on the CLEVERTex dataset compared to the other two datasets with more complex objects. Is there an explanation for this discrepancy across datasets?

**Limitations:**

yes

**Strengths And Weaknesses:**

Strengths:
The work is technically sound. The paper is well written, with high clarity and an engaging style. The theoretical motivation is clearly articulated. The new benchmark offers a systematic way to evaluate models on compositional generalization. The paper also includes detailed ablation studies and provides in-depth discussion of the results. The main finding that object-centric representations are efficient in terms of sample size and computational resources is valuable for developing models under limited data or hardware constraints.

Weaknesses:
Although the datasets are visually rich, they are still synthetic (Blender-rendered), leaving it unclear how well the proposed object-centric models would generalize to real-world scene images. In humans, object-centric representations are largely invariant to visual properties, e.g., regardless objects are depicted in realistic images or line drawings. In contrast, the models in this paper are trained and tested on specific datasets, suggesting that their object-centric representations remain far from human-level representations.
In addition, the proposed models are based on slot attention, and the comparisons are made against a relatively simple control model using k-means. The work does not include comparisons with other architectures that employ object-centric representations, such as diffusion-based models or transformer-based decoders.

---

> ### Author Rebuttal · Authors · 2026-03-31
>
> We thank the reviewer for the careful reading and the positive assessment. We are particularly glad the reviewer finds the work technically sound and the benchmark valuable. We address the weaknesses and questions below.
>
> ### W1: Only synthetic (Blender-rendered) datasets
>
> We agree that synthetic datasets do not fully match real-world complexity. However, synthetic data is essential for this type of controlled study.
>
> The goal of our benchmark is to precisely hold out 20% of object-property combinations for compositional testing, which requires knowing *exactly* which combinations exist in the world and which appear in training. This is infeasible with natural images, where attribute co-occurrence statistics cannot be fully controlled. Most prior work on object-property compositional generalization relies on the same approach for this reason (Kim et al. 2024, Montero et al. 2024), while, importantly, our synthetic worlds are visually more complex. We agree that studying real-world settings remains a valuable direction and will discuss this explicitly in the limitations section.
>
>
>
> ### W2: Models limited to Slot Attention; no diffusion-based or transformer-decoder OC
>
> We agree that the OC literature includes a range of architectures beyond Slot Attention. However, we intentionally restrict to SA-based models for two reasons.
>
> First, the goal of the study is not to propose a novel OC architecture but to measure the isolated effect of the OC inductive bias under matched backbone, capacity, and compute conditions. The aim is to maintain a controlled setup so that any gains can be attributed to the structural representation bias rather than to differences in training objectives, model capacity, or data. Adding architecturally diverse OC models would introduce additional factors of variation that would need to be accounted for.
>
> Second, SA with an MLP decoder is a simple and well-performing OC approach that does not require additional pretrained models (e.g., a pretrained diffusion decoder) and serves as a strong, widely used default (Seitzer et al. 2023, Mamaghan et al. 2024, Didolkar et al. 2025). Our comparison already goes beyond SA alone: we evaluate cross-attention (CA) resizing as an alternative learned bottleneck at the same token count (7 tokens), trained jointly with the downstream model. CA consistently underperforms DINOSAURv2 across all settings (Table 1, Section 4.2). We also experimented with replacing the SA bottleneck with a cross-attention module in the style of DINOSAURv2, which yielded substantially worse results. These experiments confirm that the SA slot-competition mechanism, not simply the number of output tokens or the training stage, is the key factor (Wu et al. 2023).
>
>
>
> ### Q1: Largest performance gains of DINOSAURv2 on CLEVRTex
>
> We agree that this is an interesting observation. To potentially rule out noise as a factor, we repeat the CLEVRTex DINOv2 vs DINOSAURv2 comparison for two additional seeds and report mean  ± std across the three seeds below. The performance difference on CLEVRTex, especially in the medium setting, is consistently large, ruling out noise as the main factor. See the response to reviewer dVeu Q2, which includes results on an additional task (property prediction), for a possible explanation.
>
> #### COOD Accuracy (mean±std, 3 seeds) - TF 2
>
> |            | CLEVRTex easy   | CLEVRTex medium  | CLEVRTex hard  |
> |:-----------|:----------------|:------------------|:----------------|
> | DINOSAURv2 | 76.3±0.9        | 71.3±0.1          | 55.1±0.7        |
> | DINOv2     | 69.9±0.3        | 59.5±0.9          | 49.6±0.3        |
>
>
> #### COOD Accuracy (mean±std, 3 seeds) - TF 5
>
> |            | CLEVRTex easy   | CLEVRTex medium  | CLEVRTex hard  |
> |:-----------|:----------------|:------------------|:----------------|
> | DINOSAURv2 | 82.9±0.6        | 72.9±0.4          | 55.2±0.3        |
> | DINOv2     | 84.2±1.4        | 70.5±1.3          | 54.8±0.7        |
>
>
> ### References
> Didolkar et al. (2025). On the transfer of object-centric representation learning. ICLR.
>
> Kim et al. (2024). Imagine the unseen world: a benchmark for systematic generalization in visual world models. NeurIPS.
>
> Mamaghan et al. (2024). Exploring the Effectiveness of Object-Centric Representations in Visual Question Answering: Comparative Insights with Foundation Models. ICLR.
>
> Montero et al. (2024). Successes and limitations of object-centric models at compositional generalisation. arXiv.
>
> Seitzer et al. (2023). Bridging the Gap to Real-World Object-Centric Learning. ICLR.
>
> Wu et al. (2023). Inverted-attention transformers can learn object representations: Insights from slot attention. Causal Representation Learning Workshop NeurIPS.

---

> > ### Author Rebuttal · Reviewer_GefW · 2026-04-03
> >
> > Thank you for the comprehensive rebuttal. However, my primary concern remains: the absence of evaluation beyond synthetic datasets, and whether findings from these datasets generalize to data that reflects real-world complexity.

---

> > > ### Author Response · Authors · 2026-04-04
> > >
> > > We thank the reviewer for their careful consideration and for appreciating the comprehensive rebuttal. We agree that evaluation on real-world images is a meaningful and important direction, and we acknowledge this as a genuine limitation of our work.
> > >
> > > At the same time, we briefly reiterate why extending to real-world data is not straightforward: our benchmark requires precisely holding out a controlled subset of object-property combinations, which demands full knowledge of which combinations exist in the world and which appear during training. Natural images do not permit this level of control, as attribute co-occurrence is governed by real-world physics and dataset curation biases. We will make this more explicit in the limitations section of the revised paper, and agree that real-world extensions are a valuable but non-trivial direction for future work.

---

### Decision · Program_Chairs · 2026-04-30

**Decision:**

Accept (regular)

**Comment:**

This study was highly evaluated for its rigorous and systematic experiments, which demonstrate the effectiveness of object-centric representations for compositional generalization, particularly under limited data diversity and computational resources.

While all reviewers shared common concerns regarding the evaluation being limited to synthetic data and VQA tasks, the lack of methodological novelty, and the uncertainty of scaling advantages, the majority of the reviewers found the empirical insights provided by the work to be of significant value. Although opinions initially diverged on whether to prioritize these limitations or the quality of the experimental findings, the skeptical reviewer softened their stance following the rebuttal process.

The AC concludes that the detailed analysis provided under constrained conditions offers valuable guidance for future research in the field, outweighing the noted concerns, and therefore recommends the paper for acceptance.